# Verified Uncertainty Calibration

**Ananya Kumar, Percy Liang, Tengyu Ma**
Department of Computer Science
Stanford University
`{ananya, pliang, tengyuma}@cs.stanford.edu`

## Abstract

Applications such as weather forecasting and personalized medicine demand models that output calibrated probability estimates—those representative of the true likelihood of a prediction. Most models are not calibrated out of the box but are recalibrated by post-processing model outputs. We find in this work that popular recalibration methods like Platt scaling and temperature scaling are (i) less calibrated than reported, and (ii) current techniques cannot estimate how miscalibrated they are. An alternative method, histogram binning, has measurable calibration error but is sample inefficient—it requires $O(B/\epsilon^2)$ samples, compared to $O(1/\epsilon^2)$ for scaling methods, where $B$ is the number of distinct probabilities the model can output. To get the best of both worlds, we introduce *the scaling-binning calibrator*, which first fits a parametric function to reduce variance and then bins the function values to actually ensure calibration. This requires only $O(1/\epsilon^2 + B)$ samples. Next, we show that we can estimate a model's calibration error more accurately using an estimator from the meteorological community—or equivalently measure its calibration error with fewer samples ($O(\sqrt{B})$ instead of $O(B)$). We validate our approach with multiclass calibration experiments on CIFAR-10 and ImageNet, where we obtain a 35% lower calibration error than histogram binning and, unlike scaling methods, guarantees on true calibration. We implement all these methods in a Python library: `https://pypi.org/project/uncertainty-calibration`

## 1 Introduction

The probability that a system outputs for an event should reflect the true frequency of that event: if an automated diagnosis system says 1,000 patients have cancer with probability 0.1, approximately 100 of them should indeed have cancer. In this case, we say the model is uncertainty calibrated. The importance of this notion of calibration has been emphasized in personalized medicine [1], meteorological forecasting [2, 3, 4, 5, 6] and natural language processing applications [7, 8]. As most modern machine learning models, such as neural networks, do not output calibrated probabilities out of the box [9, 10, 11], researchers use recalibration methods that take the output of an uncalibrated model, and transform it into a calibrated probability. *Scaling* approaches for recalibration—Platt scaling [12], isotonic regression [13], and temperature scaling [9]—are widely used and require very few samples, but do they actually produce calibrated probabilities?

*We discover that these methods are less calibrated than reported.* Past work approximates a model's calibration error using a finite set of bins. We show that by using more bins, we can uncover a higher calibration error for models on CIFAR-10 and ImageNet. We show that a fundamental limitation with approaches that output a continuous range of probabilities is that their true calibration error is unmeasurable with a finite number of bins (Example 3.2).

An alternative approach, histogram binning [10], outputs probabilities from a finite set. Histogram binning can produce a model that is calibrated, and unlike scaling methods we can measure its calibration error, but it is sample inefficient. In particular, the number of samples required to calibrate

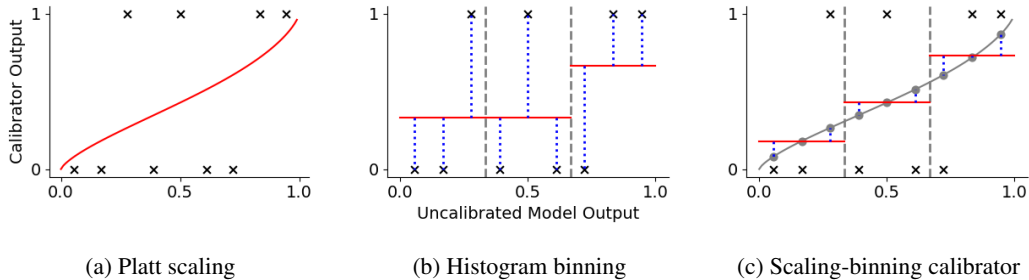

(a) Platt scaling       (b) Histogram binning       (c) Scaling-binning calibrator

Figure 1: Visualization of the three recalibration approaches. The black crosses are the ground truth labels, and the red lines are the output of the recalibration methods. Platt Scaling (Figure 1a) fits a function to the recalibration data, but its calibration error is not measurable. Histogram binning (Figure 1b) outputs the average label in each bin. The scaling-binning calibrator (Figure 1c) fits a function $g \in \mathcal{G}$ to the recalibration data and then *takes the average of the function values (the gray circles)* in each bin. The function values have lower variance than the labels, as visualized by the blue dotted lines, which is why our approach has lower variance.

scales linearly with the number of distinct probabilities the model can output, $B$ [14], which can be large particularly in the multiclass setting where $B$ typically scales with the number of classes. Recalibration sample efficiency is crucial—we often want to recalibrate our models in the presence of domain shift [15] or recalibrate a model trained on simulated data, and may have access to only a small labeled dataset from the target domain.

To get the sample efficiency of Platt scaling and the verification guarantees of histogram binning, *we propose the scaling-binning calibrator* (Figure 1c). Like scaling methods, we fit a simple function $g \in \mathcal{G}$ to the recalibration dataset. We then bin the input space so that an equal number of inputs land in each bin. In each bin, we output the average of the $g$ values in that bin—these are the gray circles in Figure 1c. In contrast, histogram binning outputs the average of the label values in each bin (Figure 1b). The motivation behind our method is that the $g$ values in each bin are in a narrower range than the label values, so when we take the average we incur lower estimation error. If $\mathcal{G}$ is well chosen, our method requires $O(\frac{1}{\epsilon^2} + B)$ samples to achieve calibration error $\epsilon$ instead of $O(\frac{B}{\epsilon^2})$ samples for histogram binning, where $B$ is the number of model outputs (Theorem 4.1). Note that in prior work, binning the outputs of a function was used for evaluation and without any guarantees, whereas in our case it is used for the method itself, and we show improved sample complexity.

We run multiclass calibration experiments on CIFAR-10 [16] and ImageNet [17]. The scaling-binning calibrator achieves a lower calibration error than histogram binning, while allowing us to measure the true calibration error unlike for scaling methods. We get a *35% lower calibration error on CIFAR-10* and a *5x lower calibration error on ImageNet* than histogram binning for $B = 100$.

Finally, we show how to estimate the calibration error of models more accurately. Prior work in machine learning [7, 9, 15, 18, 19] directly estimates each term in the calibration error from samples (Definition 5.1). The sample complexity of this plugin estimator scales linearly with $B$. A debiased estimator introduced in the meteorological literature [20, 21] reduces the bias of the plugin estimator; *we prove that it achieves sample complexity that scales with $\sqrt{B}$* by leveraging error cancellations across bins. Experiments on CIFAR-10 and ImageNet confirm that the debiased estimator measures the calibration error more accurately.

## 2 Setup and background

### 2.1 Binary classification

Let $\mathcal{X}$ be the input space and $\mathcal{Y}$ be the label space where $\mathcal{Y} = \{0, 1\}$ for binary classification. Let $X \in \mathcal{X}$ and $Y \in \mathcal{Y}$ be random variables denoting the input and label, given by an unknown joint distribution $P$. As usual, expectations are taken over all random variables.

Suppose we have a model $f : \mathcal{X} \to [0,1]$ where the (possibly uncalibrated) output of the model represents the model's confidence that the label is 1. The calibration error examines the difference between the model's probability and the true probability given the model's output:

**Definition 2.1** (Calibration error). *The calibration error of $f : \mathcal{X} \to [0,1]$ is given by:*

$$\mathrm{CE}(f) = \left( \mathbb{E}\left[ \, |f(X) - \mathbb{E}[Y \mid f(X)]|^2 \, \right] \right)^{1/2} \tag{1}$$

If $\mathrm{CE}(f) = 0$ then $f$ is perfectly calibrated. This notion of calibration error is most commonly used [2, 3, 4, 7, 15, 18, 19, 20]. Replacing the 2s in the above definition by $p \geq 1$ we get the $\ell_p$ calibration error—the $\ell_1$ and $\ell_\infty$ calibration errors are also used in the literature [9, 22, 23]. In addition to CE, we also deal with the $\ell_1$ calibration error (known as ECE) in Sections 3 and 5.

Calibration alone is not sufficient: consider an image dataset containing $50\%$ dogs and $50\%$ cats. If $f$ outputs $0.5$ on all inputs, $f$ is calibrated but not very useful. We often also wish to minimize the mean-squared error—also known as the Brier score—subject to a calibration budget [5, 24].

**Definition 2.2.** *The mean-squared error of $f : \mathcal{X} \to [0,1]$ is given by* $\mathrm{MSE}(f) = \mathbb{E}[(f(X) - Y)^2]$.

Note that MSE and CE are not orthogonal and MSE $= 0$ implies perfect calibration; in fact the MSE is the sum of the squared calibration error and a "sharpness" term [2, 4, 18].

## 2.2 Multiclass classification

While calibration in binary classification is well-studied, it's less clear what to do for multiclass, where multiple definitions abound, differing in their strengths. In the multiclass setting, $\mathcal{Y} = [K] = \{1, \ldots, K\}$ and $f : \mathcal{X} \to [0,1]^K$ outputs a confidence measure for each class in $[K]$.

**Definition 2.3** (Top-label calibration error). *The top-label calibration error examines the difference between the model's probability for its top prediction and the true probability of that prediction given the model's output:*

$$\mathrm{TCE}(f) = \left( \mathbb{E}\left[ \left( \mathbb{P}\big(Y = \underset{j \in [K]}{\arg\max}\, f(X)_j \mid \underset{j \in [K]}{\max}\, f(X)_j\big) - \underset{j \in [K]}{\max}\, f(X)_j \right)^2 \right] \right)^{1/2} \tag{2}$$

We would often like the model to be calibrated on less likely predictions as well—imagine that a medical diagnosis system says there is a $50\%$ chance a patient has a benign tumor, a $10\%$ chance she has an aggressive form of cancer, and a $40\%$ chance she has one of a long list of other conditions. We would like the model to be calibrated on all of these predictions so we define the marginal calibration error which examines, *for each class*, the difference between the model's probability and the true probability of that class given the model's output.

**Definition 2.4** (Marginal calibration error). *Let $w_k \in [0,1]$ denote how important calibrating class $k$ is, where $w_k = 1/k$ if all classes are equally important. The marginal calibration error is:*

$$\mathrm{MCE}(f) = \left( \sum_{k=1}^{K} w_k \mathbb{E}\big[(f(X)_k - \mathbb{P}(Y = k \mid f(X)_k))^2\big] \right)^{1/2} \tag{3}$$

Prior works [9, 15, 19] propose methods for multiclass calibration but only measure top-label calibration—[23] and concurrent work to ours [25] define similar per-class calibration metrics where temperature scaling [9] is worse than vector scaling despite having better top-label calibration.

For notational simplicity, our theory focuses on the binary classification setting. We can transform top-label calibration into a binary calibration problem—the model outputs a probability corresponding to its top prediction, and the label represents whether the model gets it correct or not. Marginal calibration can be transformed into $K$ one-vs-all binary calibration problems where for each $k \in [K]$ the model outputs the probability associated with the $k$-th class, and the label represents whether the correct class is $k$ [13]. We consider both top-label calibration and marginal calibration in our experiments. Other notions of multiclass calibration include joint calibration (which requires the entire probability *vector* to be calibrated) [2, 6] and event-pooled calibration [18].

## 2.3 Recalibration

Since most machine learning models do not output calibrated probabilities out of the box [9, 10] recalibration methods take the output of an uncalibrated model, and transform it into a calibrated probability. That is, given a trained model $f : \mathcal{X} \to [0, 1]$, let $Z = f(X)$. We are given recalibration data $T = \{(z_i, y_i)\}_{i=1}^n$ independently sampled from $P(Z, Y)$, and we wish to learn a recalibrator $g : [0, 1] \to [0, 1]$ such that $g \circ f$ is well-calibrated.

*Scaling methods*, for example Platt scaling [12], output a function $g = \arg\min_{g \in \mathcal{G}} \sum_{(z,y) \in T} \ell(g(z), y)$, where $\mathcal{G}$ is a model family, $g \in \mathcal{G}$ is differentiable, and $\ell$ is a loss function, for example the log-loss or mean-squared error. The advantage of such methods is that they converge very quickly since they only fit a small number of parameters.

*Histogram binning* first constructs a set of bins (intervals) that partitions $[0, 1]$, formalized below.

**Definition 2.5** (Binning schemes). *A binning scheme $\mathcal{B}$ of size $B$ is a set of $B$ intervals $I_1, \ldots, I_B$ that partitions $[0, 1]$. Given $z \in [0, 1]$, let $\beta(z) = j$, where $j$ is the interval that $z$ lands in ($z \in I_j$).*

The bins are typically chosen such that either $I_1 = [0, \frac{1}{B}], I_2 = (\frac{1}{B}, \frac{2}{B}], \ldots, I_B = (\frac{B-1}{B}, 1]$ (equal width binning) [9] or so that each bin contains an equal number of $z_i$ values in the recalibration data (uniform mass binning) [10]. Histogram binning then outputs the average $y_i$ value in each bin.

# 3 Is Platt scaling calibrated?

In this section, we show that methods like Platt scaling and temperature scaling are (i) less calibrated than reported and (ii) it is difficult to tell how miscalibrated they are. That is we show, both theoretically and with experiments on CIFAR-10 and ImageNet, why the calibration error of models that output a continuous range of values is *underestimated*. We defer proofs to Appendix B.

The key to estimating the calibration error is estimating the conditional expectation $\mathbb{E}[Y \mid f(X)]$. If $f(X)$ is continuous, without smoothness assumptions on $\mathbb{E}[Y \mid f(X)]$ (that cannot be verified in practice), this is impossible. This is analogous to the difficulty of measuring the mutual information between two continuous signals [26].

To approximate the calibration error, prior work bins the output of $f$ into $B$ intervals. The calibration error in each bin is estimated as the difference between the average value of $f(X)$ and $Y$ in that bin. Note that the binning here is for evaluation only, whereas in histogram binning, it is used for the recalibration method itself. We formalize the notion of this binned calibration error below.

**Definition 3.1.** *The binned version of $f$ outputs the average value of $f$ in each bin $I_j$:*

$$f_\mathcal{B}(x) = \mathbb{E}[f(X) \mid f(X) \in I_j] \qquad where \; x \in I_j \tag{4}$$

Given $\mathcal{B}$, the binned calibration error of $f$ is simply the calibration error of $f_\mathcal{B}$. A simple example shows that using binning to estimate the calibration error can severely underestimate the true calibration error.

**Example 3.2.** *For any binning scheme $\mathcal{B}$, and continuous bijective function $f : [0, 1] \to [0, 1]$, there exists a distribution $P$ over $\mathcal{X}, \mathcal{Y}$ s.t. $\mathrm{CE}(f_\mathcal{B}) = 0$ but $\mathrm{CE}(f) \geq 0.49$. Note that for all $f$, $0 \leq \mathrm{CE}(f) \leq 1$.*

The intuition of the construction is that in each interval $I_j$ in $\mathcal{B}$, the model could underestimate the true probability $\mathbb{E}[Y \mid f(X)]$ half the time, and overestimate the probability half the time. So if we average over the entire bin the model appears to be calibrated, even though it is very uncalibrated. The formal proof is in Appendix B, and holds for arbitrary $\ell_p$ calibration errors including the ECE.

Next, we show that given a function $f$, its binned version always has lower calibration error. The proof, in Appendix B, is by Jensen's inequality. Intuitively, averaging a model's prediction within a bin allows errors at different parts of the bin to cancel out with each other. This result is similar to Theorem 2 in recent work [27], and holds for arbitrary $\ell_p$ calibration errors including the ECE.

**Proposition 3.3** (Binning underestimates error). *Given any binning scheme $\mathcal{B}$ and model $f : \mathcal{X} \to [0, 1]$, we have:*

$$\mathrm{CE}(f_\mathcal{B}) \leq \mathrm{CE}(f).$$

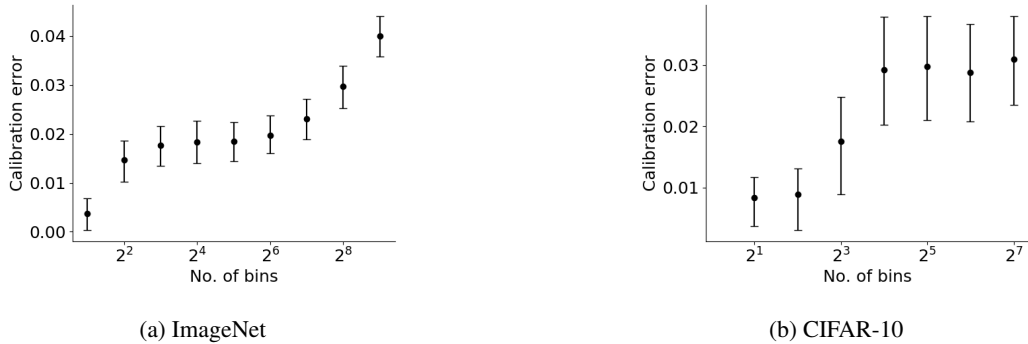

|  (a) ImageNet | (b) CIFAR-10 |

Figure 2: Binned calibration errors of a recalibrated VGG-net model on CIFAR-10 and ImageNet with 90% confidence intervals. The binned calibration error increases as we increase the number of bins. This suggests that binning cannot be reliably used to measure the true calibration error.

## 3.1 Experiments

Our experiments on ImageNet and CIFAR-10 suggest that previous work reports numbers which are lower than the actual calibration error of their models. Recall that binning lower bounds the calibration error. We cannot compute the actual calibration error but if we use a 'finer' set of bins then we get a tighter lower bound on the calibration error.

As in [9], our model's objective was to output the top predicted class and a confidence score associated with the prediction. For ImageNet, we started with a trained VGG16 model with an accuracy of 64.3%. We split the validation set into 3 sets of sizes $(20000, 5000, 25000)$. We used the first set of data to recalibrate the model using Platt scaling, the second to select the binning scheme $\mathcal{B}$ so that each bin contains an equal number of points, and the third to measure the binned calibration error . We calculated 90% confidence intervals for the binned calibration error using 1,000 bootstrap resamples and performed the same experiment with varying numbers of bins.

Figure 2a shows that as we increase the number of bins on ImageNet, the measured calibration error is higher and this is statistically significant. For example, if we use 15 bins as in [9], we would think the calibration error is around 0.02 when in reality the calibration error is at least twice as high. Figure 2b shows similar findings for CIFAR-10, and in Appendix C we show that our findings hold even if we use the $\ell_1$ calibration error (ECE) and alternative binning strategies.

## 4 The scaling-binning calibrator

Section 3 shows that the problem with scaling methods is we cannot estimate their calibration error. The upside of scaling methods is that if the function family has at least one function that can achieve calibration error $\epsilon$, they require $O(1/\epsilon^2)$ samples to reach calibration error $\epsilon$, while histogram binning requires $O(B/\epsilon^2)$ samples. Can we devise a method that is sample efficient to calibrate and one where it's possible to estimate the calibration error? To achieve this, we propose the scaling-binning calibrator (Figure 1c) where we first fit a scaling function, and then bin the outputs of the scaling function.

### 4.1 Algorithm

We split the recalibration data $T$ of size $n$ into 3 sets: $T_1, T_2, T_3$. The scaling-binning calibrator, illustrated in Figure 1, outputs $\hat{g_{\mathcal{B}}}$ such that $\hat{g_{\mathcal{B}}} \circ f$ has low calibration error:

**Step 1 (Function fitting):** Select $g = \arg\min_{g \in \mathcal{G}} \sum_{(z,y) \in T_1} (y - g(z))^2$.

**Step 2 (Binning scheme construction):** We choose the bins so that an equal number of $g(z_i)$ in $T_2$ land in each bin $I_j$ for each $j \in \{1, \ldots, B\}$—this uniform-mass binning scheme [10] as opposed to equal-width binning [9] is essential for being able to estimate the calibration error in Section 5.

**Step 3 (Discretization):** Discretize $g$, by outputting the average $g$ value in each bin—these are the gray circles in Figure 1c. Let $\mu(S) = \frac{1}{|S|} \sum_{s \in S} s$ denote the mean of a set of values $S$. Let $\hat{\mu}[j] = \mu(\{g(z_i) \mid g(z_i) \in I_j \wedge (z_i, y_i) \in T_3\})$ be the mean of the $g(z_i)$ values that landed in the $j$-th bin. Recall that if $z \in I_j$, $\beta(z) = j$ is the interval z lands in. Then we set $\hat{g}_{\mathcal{B}}(z) = \hat{\mu}[\beta(g(z))]$—that is we simply output the mean value in the bin that $g(z)$ falls in.

## 4.2 Analysis

We now show that the scaling-binning calibrator requires $O(B + 1/\epsilon^2)$ samples to calibrate, and in Section 5 we show that we can efficiently measure its calibration error. For the main theorem, we make some standard regularity assumptions on $\mathcal{G}$ which we formalize in Appendix D. Our result is a generalization result—we show that if $\mathcal{G}$ contains some $g^*$ with low calibration error, then our method is *at least* almost as well-calibrated as $g^*$ given sufficiently many samples.

**Theorem 4.1** (Calibration bound). *Assume regularity conditions on $\mathcal{G}$ (finite parameters, injectivity, Lipschitz-continuity, consistency, twice differentiability). Given $\delta \in (0, 1)$, there is a constant $c$ such that for all $B, \epsilon > 0$, with $n \geq c\left(B \log B + \frac{\log B}{\epsilon^2}\right)$ samples, the scaling-binning calibrator finds $\hat{g}_{\mathcal{B}}$ with $(\mathrm{CE}(\hat{g}_{\mathcal{B}}))^2 \leq 2 \min_{g \in \mathcal{G}} (\mathrm{CE}(g))^2 + \epsilon^2$, with probability $\geq 1 - \delta$.*

Note that our method can potentially be better calibrated than $g^*$, because we bin the outputs of the scaling function, which reduces its calibration error (Proposition 3.3). While binning worsens the sharpness and can increase the mean-squared error of the model, in Proposition D.4 we show that if we use many bins, binning the outputs cannot increase the mean-squared error by much.

We prove Theorem 4.1 in Appendix D but give a sketch here. Step 1 of our algorithm is Platt scaling, which simply fits a function $g$ to the data—standard results in asymptotic statistics show that $g$ converges in $O(\frac{1}{\epsilon^2})$ samples.

Step 3, where we bin the outputs of $g$, is the main step of the algorithm. If we had infinite data, Proposition 3.3 showed that the binned version $g_{\mathcal{B}}$ has lower calibration error than $g$, so we would be done. However we do not have infinite data—the core of our proof is to show that the empirically binned $\hat{g}_{\mathcal{B}}$ is within $\epsilon$ of $g_{\mathcal{B}}$ in $O(B + \frac{1}{\epsilon^2})$ samples, instead of the $O(B + \frac{B}{\epsilon^2})$ samples required by histogram binning. The intuition is in Figure 1—the $g(z_i)$ values in each bin (gray circles in Figure 1c) are in a narrower range than the $y_i$ values (black crosses in Figure 1b) and thus have lower variance so when we take the average we incur less estimation error. The perhaps surprising part is that we are estimating $B$ numbers with $\widetilde{O}(1/\epsilon^2)$ samples. In fact, there may be a small number of bins where the $g(z_i)$ values are not in a narrow range, but our proof still shows that the overall estimation error is small.

Our uniform-mass binning scheme allows us to estimate the calibration error efficiently (see Section 5), unlike for scaling methods where we cannot estimate the calibration error (Section 3). Recall that we chose our bins so that each bin has an equal proportion of points in the recalibration set. Lemma 4.3 shows that this property approximately holds in the population as well. This allows us to estimate the calibration error efficiently (Theorem 5.4).

**Definition 4.2** (Well-balanced binning). *Given a binning scheme $\mathcal{B}$ of size $B$, and $\alpha \geq 1$. We say $\mathcal{B}$ is $\alpha$-well-balanced if for all $j$,*

$$\frac{1}{\alpha B} \leq \mathbb{P}(Z \in I_j) \leq \frac{\alpha}{B}$$

**Lemma 4.3.** *For universal constant $c$, if $n \geq cB \log \frac{B}{\delta}$, with probability at least $1 - \delta$, the binning scheme $\mathcal{B}$ we chose is 2-well-balanced.*

While the way we choose bins is not novel [10], we believe the guarantees around it are—not all binning schemes in the literature allow us to efficiently estimate the calibration error; for example, the binning scheme in [9] does not. Our proof of Lemma 4.3 is in Appendix D. We use a discretization argument to prove the result—this gives a tighter bound than applying Chernoff bounds or a standard VC dimension argument which would tell us we need $O(B^2 \log \frac{B}{\delta})$ samples.

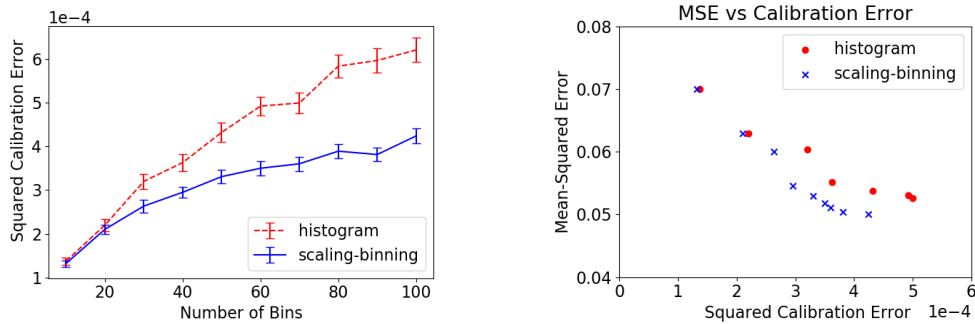

(a) Effect of number of bins on squared calibration error.　　(b) Tradeoff between calibration and MSE.

Figure 3: (**Left**) Recalibrating using 1,000 data points on CIFAR-10, the scaling-binning calibrator achieves lower squared calibration error than histogram binning, especially when the number of bins $B$ is large. (**Right**) For a fixed calibration error, the scaling-binning calibrator allows us to use more bins. This results in models with more predictive power which can be measured by the mean-squared error. Note the vertical axis range is $[0.04, 0.08]$ to zoom into the relevant region.

## 4.3   Experiments

Our experiments on CIFAR-10 and ImageNet show that in the low-data regime, for example when we use $\leq 1000$ data points to recalibrate, the scaling-binning calibrator produces models with much lower calibration error than histogram binning. The uncalibrated model outputs a confidence score associated with each class. We recalibrated each class separately as in [13], using $B$ bins per class, and evaluated calibration using the marginal calibration error (Definition 2.4).

We describe our experimental protocol for CIFAR-10. The CIFAR-10 validation set has 10,000 data points. We sampled, with replacement, a recalibration set of 1,000 points. We ran either the scaling-binning calibrator (we fit a sigmoid in the function fitting step) or histogram binning and measured the marginal calibration error on the entire set of 10K points. We repeated this entire procedure 100 times and computed mean and 90% confidence intervals, and we repeated this varying the number of bins $B$. Figure 3a shows that the scaling-binning calibrator produces models with lower calibration error, for example 35% lower calibration error when we use 100 bins per class.

Using more bins allows a model to produce more fine-grained predictions, e.g. [20] use $B = 51$ bins, which improves the quality of predictions as measured by the mean-squared error—Figure 3b shows that our method achieves better mean-squared errors for any given calibration constraint. More concretely, the figure shows a scatter plot of the mean-squared error and squared calibration error for histogram binning and the scaling-binning calibrator when we vary the number of bins. For example, if we want our models to have a calibration error $\leq 0.02 = 2\%$ we get a 9% lower mean-squared error. In Appendix E we show that we get *5x lower top-label calibration error on ImageNet*, and give further experiment details.

**Validating theoretical bounds**: In Appendix E we run synthetic experiments to validate the bound in Theorem 4.1. In particular, we show that if we fix the number of samples $n$, and vary the number of bins $B$, the squared calibration error for the scaling-binning calibrator is nearly constant but for histogram binning increases nearly linearly with $B$. For both methods, the squared calibration error decreases approximately as $1/n$—that is when we double the number of samples the squared calibration error halves.

## 5   Verifying calibration

Before deploying our model we would like to check that it has calibration error below some desired threshold $\mathcal{E}$. In this section we show that we can accurately estimate the calibration error of binned models, if the binning scheme is 2-well-balanced. Recent work in machine learning uses a plugin estimate for each term in the calibration error [7, 15, 18, 19]. Older work in meteorology [20, 21] notices that this is a biased estimate, and proposes a *debiased* estimator that subtracts off an

approximate correction term to reduce the bias. Our contribution is to show that the debiased estimator is more accurate: while the plugin estimator requires samples proportional to $B$ to estimate the calibration error, the debiased estimator requires samples proportional to $\sqrt{B}$. Note that we show an *improved sample complexity*—prior work only showed that the naive estimator is biased. In Appendix G we also propose a way to debias the $\ell_1$ calibration error (ECE), and show that we can estimate the ECE more accurately on CIFAR-10 and ImageNet.

Suppose we wish to measure the squared calibration error $\mathcal{E}^2$ of a binned model $f : \mathcal{X} \to S$ where $S \subseteq [0, 1]$ and $|S| = B$. Suppose we get an evaluation set $T_n = \{(x_1, y_1), \ldots, (x_n, y_n)\}$. Past work typically estimates the calibration error by directly estimating each term from samples:

**Definition 5.1** (Plugin estimator). *Let $L_s$ denote the $y_j$ values where the model outputs $s$: $L_s = \{y_j \mid (x_j, y_j) \in T_n \wedge f(x_j) = s\}$. Let $\hat{p}_s$ be the estimated probability of $f$ outputting $s$: $\hat{p}_s = \frac{|L_s|}{n}$.*

*Let $\hat{y}_s$ be the empirical average of $Y$ when the model outputs $s$: $\hat{y}_s = \sum_{y \in L_s} \frac{y}{|L_s|}$.*

*The plugin estimate for the squared calibration error is the weighted squared difference between $\hat{y}_s$ and $s$:*

$$\hat{\mathcal{E}}_{\mathrm{pl}}^2 = \sum_{s \in S} \hat{p}_s (s - \hat{y}_s)^2$$

Alternatively, [20, 21] propose to subtract an approximation of the bias from the estimate:

**Definition 5.2** (Debiased estimator). *The debiased estimator for the squared calibration error is:*

$$\hat{\mathcal{E}}_{\mathrm{db}}^2 = \sum_{s \in S} \hat{p}_s \left[ (s - \hat{y}_s)^2 - \frac{\hat{y}_s(1 - \hat{y}_s)}{\hat{p}_s n - 1} \right]$$

We are interested in analyzing the number of samples required to estimate the calibration error within a constant multiplicative factor, that is to give an estimate $\hat{\mathcal{E}}^2$ such that $|\hat{\mathcal{E}}^2 - \mathcal{E}^2| \leq \frac{1}{2} \mathcal{E}^2$ (where $\frac{1}{2}$ can be replaced by any constant $r$ with $0 < r < 1$). Our main result is that the plugin estimator requires $\widetilde{O}(\frac{B}{\mathcal{E}^2})$ samples (Theorem 5.3) while the debiased estimator requires $\widetilde{O}(\frac{\sqrt{B}}{\mathcal{E}^2})$ samples (Theorem 5.4).

**Theorem 5.3** (Plugin estimator bound). *Suppose we have a binned model with squared calibration error $\mathcal{E}^2$, where the binning scheme is 2-well-balanced, that is for all $s \in S$, $\mathbb{P}(f(X) = s) \geq \frac{1}{2B}$. [1] If $n \geq c \frac{B}{\mathcal{E}^2} \log \frac{B}{\delta}$ for some universal constant $c$, then for the plugin estimator, we have $\frac{1}{2} \mathcal{E}^2 \leq \hat{\mathcal{E}}_{\mathrm{pl}}^2 \leq \frac{3}{2} \mathcal{E}^2$ with probability at least $1 - \delta$.*

**Theorem 5.4** (Debiased estimator bound). *Suppose we have a binned model with squared calibration error $\mathcal{E}^2$ and for all $s \in S$, $\mathbb{P}(f(X) = s) \geq \frac{1}{2B}$. If $n \geq c \frac{\sqrt{B}}{\mathcal{E}^2} \log \frac{B}{\delta}$ for some universal constant $c$ then for the debiased estimator, we have $\frac{1}{2} \mathcal{E}^2 \leq \hat{\mathcal{E}}_{\mathrm{db}}^2 \leq \frac{3}{2} \mathcal{E}^2$ with probability at least $1 - \delta$.*

The proofs of both theorems is in Appendix F. The idea is that for the plugin estimator, each term in the sum has bias $1/n$. These biases accumulate, giving total bias $B/n$. The debiased estimator has much lower bias and the estimation variance cancels across bins—this intuition is captured in Lemma F.8 which requires careful conditioning to make the argument go through.

## 5.1 Experiments

We run a multiclass marginal calibration experiment on CIFAR-10 which suggests that the debiased estimator produces better estimates of the calibration error than the plugin estimator. We split the validation set of size 10,000 into two sets $S_C$ and $S_E$ of sizes 3,000 and 7,000 respectively. We use $S_C$ to re-calibrate and discretize a trained VGG-16 model. We calibrate each of the $K = 10$ classes seprately as described in Section 2 and used $B = 100$ or $B = 10$ bins per class. For varying values of $n$, we sample $n$ points with replacement from $S_E$, and estimate the calibration error using the debiased estimator and the plugin estimator. We then compute the squared deviation of these estimates from the squared calibration error measured on the entire set $S_E$. We repeat this resampling 1,000 times to get the mean squared deviation of the estimates from the ground truth and confidence intervals. Figure 4a shows that the debiased estimates are much closer to the ground truth than the plugin estimates—the difference is especially significant when the number of samples $n$ is small or

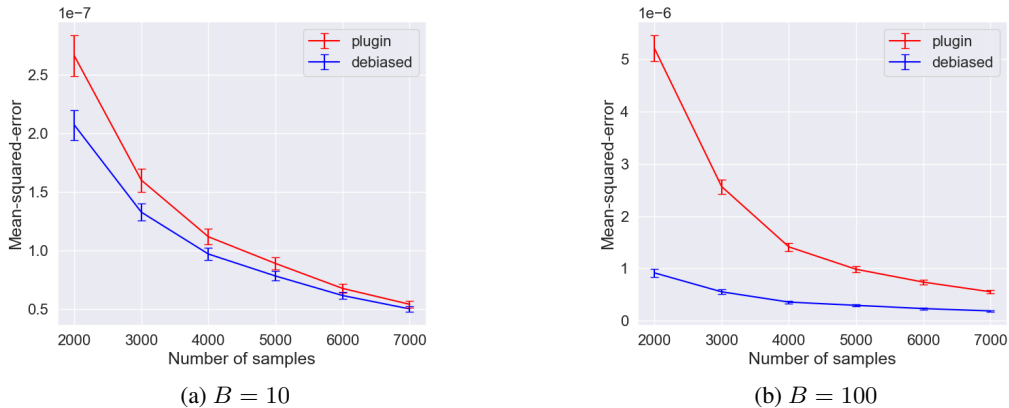

(a) $B = 10$                                                     (b) $B = 100$

Figure 4: Mean-squared errors of plugin and debiased estimators on a recalibrated VGG16 model on CIFAR-10 with $90\%$ confidence intervals (lower values better). The debiased estimator is closer to the ground truth, which corresponds to $0$ on the vertical axis, especially when $B$ is large or $n$ is small. Note that this is the MSE of the squared calibration error, not the MSE of the model in Figure 3.

the number of bins $B$ is large. Note that having a perfect estimate corresponds to $0$ on the vertical axis.

In Appendix G, we include histograms of the absolute difference between the estimates and ground truth for the plugin and debiased estimator, over the 1,000 resamples.

## 6 Related work

Calibration, including the squared calibration error, has been studied in many fields besides machine learning including meteorology [2, 3, 4, 5, 6], fairness [28, 29], healthcare [1, 30, 31, 32], reinforcement learning [33], natural language processing [7, 8], speech recognition [34], econometrics [24], and psychology [35]. Besides the calibration error, prior work also uses the Hosmer-Lemeshov test [36] and reliability diagrams [4, 37] to evaluate calibration. Concurrent work to ours [38] also notice that using the plugin calibration error estimator to test for calibration leads to rejecting well-calibrated models too often. Besides calibration, other ways of producing and quantifying uncertainties include Bayesian methods [39] and conformal prediction [40, 41].

Algorithms and analysis in density estimation typically assume the true density is $L-$Lipschitz, while in calibration applications, the calibration error of the final model should be measurable from data, without making untestable assumptions on $L$.

Bias is a common issue with statistical estimators, for example, the seminal work by Stein [42] fixes the bias of the mean-squared error. However, debiasing an estimator does not typically lead to *an improved sample complexity*, as it does in our case. Recalibration is related to (conditional) density estimation [43, 44] as the goal is to estimate $\mathbb{E}[Y \mid f(X)]$.

## 7 Conclusion

This paper makes three contributions: 1. We showed that the calibration error of continuous methods is underestimated; 2. We introduced the first method, to our knowledge, that has better sample complexity than histogram binning and has a *measurable calibration error*, giving us the best of scaling and binning methods; and 3. We showed that an alternative estimator for calibration error has better sample complexity than the plugin estimator. There are many exciting avenues for future work:

1. **Dataset shifts**: Can we maintain calibration under dataset shifts (for example, train on MNIST, but evaluate on SVHN) without labeled examples from the target dataset?

2. **Measuring calibration**: Can we come up with alternative metrics that still capture a notion of calibration, but are measurable for scaling methods?

**Reproducibility.** Our Python calibration library is available at `https://pypi.org/project/uncertainty-calibration`. All code, data, and experiments can be found on CodaLab at `https://worksheets.codalab.org/worksheets/0xb6d027ee127e422989ab9115726c5411`. Updated code can be found at `https://github.com/AnanyaKumar/verified_calibration`.

**Acknowledgements.** The authors would like to thank the Open Philantropy Project, Stanford Graduate Fellowship, and Toyota Research Institute for funding. Toyota Research Institute ("TRI") provided funds to assist the authors with their research but this article solely reflects the opinions and conclusions of its authors and not TRI or any other Toyota entity.

We are grateful to Pang Wei Koh, Chuan Guo, Anand Avati, Shengjia Zhao, Weihua Hu, Yu Bai, John Duchi, Dan Hendrycks, Jonathan Uesato, Michael Xie, Albert Gu, Aditi Raghunathan, Fereshte Khani, Stefano Ermon, Eric Nalisnick, and Pushmeet Kohli for insightful discussions. We thank the anonymous reviewers for their thorough reviews and suggestions that have improved our paper. We would also like to thank Pang Wei Koh, Yair Carmon, Albert Gu, Rachel Holladay, and Michael Xie for their inputs on our draft, and Chuan Guo for providing code snippets from their temperature scaling paper.

## Footnotes

[1]We do not need the upper bound of the 2-well-balanced property.

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
