[Supplementary Material · calibration_neurips_2019.pdf]

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

[2]Note that the refinement term can be further decomposed into resolution (also known as sharpness) and irreducible uncertainty.

[3]This is a very technical point, so at a first pass the reader may skip the following discussion.

[4]We do not need the upper bound of the 2-well-balanced property.

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

# A  Model and code details

The VGG16 model we used for ImageNet experiments is from the Keras [45] module in the TensorFlow [46] library and we used pre-trained weights supplied by the library. The VGG16 model for CIFAR-10 was obtained from an open-source implementation on GitHub [47], and we used the pre-trained weights there. We independently verified the accuracies of these models.

# B Proofs for Section 3

The results in Section 3 hold more generally for $\ell_p$-CE and not just CE. We recall the definition of $\ell_p$-CE:

**Definition B.1.** *For $p \geq 1$, the $\ell_p$ calibration error of $f : \mathcal{X} \to [0,1]$ is given by:*

$$\ell_p\text{-CE}(f) = \Big( \mathbb{E}\left[\, |f(X) - \mathbb{E}[Y \mid f(X)]|^p \,\right] \Big)^{1/p} \tag{5}$$

We now restate and prove the results in terms of $\ell_p$-CE which implies the result for CE in the paper as a special case, but also for other commonly used metrics such as the $\ell_1$ calibration error (ECE).

**Restatement of Example 3.2.** *For any binning scheme $\mathcal{B}$, $p \geq 1$, and continuous bijective function $f : [0,1] \to [0,1]$, there exists a distribution $P$ over $\mathcal{X}, \mathcal{Y}$ s.t. $\ell_p\text{-CE}(f_{\mathcal{B}}) = 0$ but $\ell_p\text{-CE}(f) \geq 0.49$. Note that for all $f$, $0 \leq \ell_p\text{-CE}(f) \leq 1$.*

*Proof.* As stated in the main text, the intuition is that in each interval $I_j$ in $\mathcal{B}$, the model could underestimate the true probability $\mathbb{E}[Y \mid f(X)]$ half the time, and overestimate the probability half the time. So if we average over the entire bin the model appears to be calibrated, even though it is very uncalibrated. The proof simply formalizes this intuition.

Since $f$ is bijective and continuous we can select data distribution $P$ s.t. $f(X) \sim \text{Uniform}[0.5 - \epsilon, 0.5 + \epsilon]$ for any $\epsilon > 0$. To see this, first note that from real analysis since $f : [0,1] \to [0,1]$ and $f$ is bijective and continuous, $f^{-1}$ is also bijective and continuous. Then we can let $X \sim f^{-1}(\text{Uniform}[0.5 - \epsilon, 0.5 + \epsilon])$ which has the desired property and has a density.

Now, consider each interval $I_j$ in binning scheme $\mathcal{B}$, and let $A_j = I_j \cap \text{Uniform}[0.5 - \epsilon, 0.5 + \epsilon]$. If $A_j = \emptyset$ then $P(f(X) \in A_j) = 0$ so we can ignore this interval (since $f(X)$ will never land in this bin). Let $p_j = \mathbb{E}[f(X) \mid f(X) \in A_j]$. Note that $\mathbb{E}[f(X) \mid f(X) \in A_j] = \mathbb{E}[f(X) \mid f(X) \in I_j]$. Since $f(X) \in [0.5 - \epsilon, 0.5 + \epsilon]$, $p_j \in [0.5 - \epsilon, 0.5 + \epsilon]$ as well. We will choose $P(Y)$ so that $Y$ is 1 whenever $f(X)$ lands in the first $p_j$ fraction of interval $A_j$, and 0 whenever $f(X)$ lands in the latter $1 - p_j$ fraction of $A_j$. Then $\mathbb{E}[Y \mid f(X) \in A_j] = p_j$, so the binned calibration error is 0. But notice that for all $s \in [0.5 - \epsilon, 0.5 + \epsilon]$, $\mathbb{E}[Y \mid f(X) = s]$ is either 0 or 1. So we have:

$$|\mathbb{E}[Y \mid f(X) = s] - s| \geq 0.5 - \epsilon$$

That is, at every point the model is actually very miscalibrated at each $s$. By taking $\epsilon$ very small, we then get that $\ell_p\text{-CE}(p) \geq 0.5 - \epsilon'$ for any $\epsilon' > 0$, which completes the proof. $\square$

**Restatement of Proposition 3.3.** *Given any binning scheme $\mathcal{B}$ and model $f : \mathcal{X} \to [0,1]$, we have:*

$$\ell_p\text{-CE}(f_{\mathcal{B}}) \leq \ell_p\text{-CE}(f).$$

*Proof.* It suffices to prove the claim for the $\ell_p^p$ error:

$$(\ell_p\text{-CE}(f_{\mathcal{B}}))^p \leq (\ell_p\text{-CE}(f))^p$$

This is because if $p > 0$ then $a \leq b \Leftrightarrow a^p \leq b^p$.

For $p \geq 1$, let $l(a,b) = (|a - b|)^p$. We note that $l$ is convex in both arguments. The proof is now a simple result of Jensen's inequality and convexity of $l$. Suppose that $\mathcal{B}$ is given by intervals $I_1, ..., I_B$. Let $Z = f(X)$—note that $Z$ is a random variable.

We can write $(\ell_p\text{-CE}(f_{\mathcal{B}}))^p$ as:

$$(\ell_p\text{-CE}(f_{\mathcal{B}}))^p = \sum_{j=1}^{B} P(Z \in I_j)\, l\Big(\mathbb{E}[Z \mid Z \in I_j], \mathbb{E}[Y \mid Z \in I_j]\Big)$$

We can write $(\ell_p\text{-CE}(f))^p$ as:

$$(\ell_p\text{-CE}(f))^p = \sum_{j=1}^{B} P(Z \in I_j)\, \mathbb{E}\Big[l\big(Z, \mathbb{E}[Y \mid Z]\big) \mid Z \in I_j\Big]$$

Fix some bin $I_j \in \mathcal{B}$. By Jensen's inequality,

$$l\Big(\mathbb{E}[Z \mid Z \in I_j], \mathbb{E}[Y \mid Z \in I_j]\Big) \le \mathbb{E}\Big[l\big(Z, \mathbb{E}[Y \mid Z]\big) \mid Z \in I_j\Big]$$

Since this inequality holds for each term in the sum, it holds for the whole sum:

$$(\ell_p\text{-CE}(f_{\mathcal{B}}))^p \le (\ell_p\text{-CE}(f))^p$$

Note that the proof also implies that finer binning schemes give a better lower bound. That is, given $\mathcal{B}'$ suppose for all $I'_j \in \mathcal{B}'$, $I'_j \subseteq I_k$ for some $I_k \in \mathcal{B}$. Then $\ell_p\text{-CE}(f_{\mathcal{B}}) \le \ell_p\text{-CE}(f_{\mathcal{B}'}) \le \ell_p\text{-CE}(f)$. This is because $f_{\mathcal{B}'}$ can be seen as a binned version of $f_{\mathcal{B}}$.

$\square$

(a) ImageNet, $\ell_1$-CE                     (b) ImageNet, $\ell_1$-CE, equal-width binning

Figure 5: Binned $\ell_1$ calibration errors of a recalibrated VGG-net model on ImageNet with 90% confidence intervals. The binned calibration error increases as we increase the number of bins. This suggests that binning cannot be reliably used to measure the $\ell_1$-CE.

(a) CIFAR-10, $\ell_1$-CE                     (b) CIFAR-10, $\ell_1$-CE, equal-width binning

Figure 6: Binned $\ell_1$ calibration errors of a recalibrated VGG-net model on CIFAR-10 with 90% confidence intervals. The results are not as conclusive here because the error bars are large, however it seems to suggest that the binned calibration error increases as we increase the number of bins.

## C  Ablations for Section 3

Here we present additional experiments for Section 3. Recall that the experiments in section 3 showed that binning underestimates the calibration error of a model—we focused on the $\ell_2$-CE and selected bins so that each bin has an equal number of data points. Figure 5a shows that binning is also unreliable at measuring the $\ell_1$-CE (ECE) on ImageNet—using more bins uncovers a higher calibration error than we would otherwise detect with fewer bins. Figure 5b shows that the same conclusion holds on ImageNet if we look at the $\ell_1$-CE *and* use an alternative approach to selecting bins used in [9] that we call *equal-width binning*. Here, the $B$ bins are selected to be $I_1 = [0, \frac{1}{B}], I_2 = (\frac{1}{B}, \frac{2}{B}], \dots, I_B = (\frac{B-1}{B}, 1]$. The experimental protocol is the same as in section 3.

We repeated both of these experiments on CIFAR-10 as well, and plot the results in Figure 6. Here the results are inconclusive because the error bars are large. This is because the CIFAR-10 dataset is smaller than ImageNet, and the accuracy of the CIFAR-10 model is 93.1%, so the calibration error that we are trying to measure is much smaller.

We provide details on the dataset split for CIFAR-10. For CIFAR-10, we used a VGG16 model and split the test set into 3 sets of size $(1000, 1000, 8000)$, where used the first set of data to recalibrate the model using Platt scaling, the second to select the binning scheme, and the third to measure the binned calibration error. As stated in the main body of the paper, for ImageNet we used a split of $(20000, 5000, 25000)$.

# D   Proofs for section 4

Our analysis of the sample complexity of the scaling-binning calibrator requires some assumptions on the function family $\mathcal{G}$:

1. (Finite parameters). Let $\mathcal{G} = \{g_\theta : [0,1] \to [0,1] \mid \theta \in A\}$ where $A \subseteq \mathbb{R}^d$ and $A$ is open.

2. (Injective). For all $g_\theta \in \mathcal{G}$ we assume $g_\theta$ is injective.

3. (Consistency). Intuitively, consistency means that given infinite data, the estimated parameters should converge to the unique optimal parameters in $A$. More formally, suppose $\theta^* = \arg\min_{\theta \in A} \text{MSE}(g_\theta)$. Then the parameters $\hat{\theta}_n$ estimated by minimizing the empirical MSE with $n$ samples in step 1 of the algorithm, converges in distribution to $\theta^*$, that is, $\hat{\theta}_n \to_D \theta^*$ as $n \to \infty$. Note that consistency inherently assumes identifiability, that there is a unique minimizer $\theta^*$ in the open set $A$.

4. (Regularity). We assume that regularity conditions in Theorem 5.23 of [48] hold, which require the loss to be twice differentiable with symmetric, non-singular Hessian, and that $g_\theta(x)$ is Lipschitz in $\theta$ for all $x$. We will also require the second derivative to be continuous.

We assume that $\mathcal{G}$ satisfies these assumptions in the rest of this section. Note that aside from injectivity, the remaining conditions are only required for the (fairly standard) analysis of step 1 of the algorithm, which says that a parametric scaling method with a small number of parameters will quickly converge to its optimal error.

## D.1   Calibration bound (Proof of Theorem 4.1)

The goal is to prove the following theorem from Section 4, which we restate:

**Restatement of Theorem 4.1.** *Assume regularity conditions on $\mathcal{G}$ (finite parameters, injectivity, Lipschitz-continuity, consistency, twice differentiability). Given $\delta \in (0,1)$, there is a constant $c$ such that for all $B, \epsilon > 0$, with $n \geq c\left(B \log B + \frac{\log B}{\epsilon^2}\right)$ samples, the scaling-binning calibrator finds $\hat{g}_\mathcal{B}$ with $(\text{CE}(\hat{g}_\mathcal{B}))^2 \leq 2\min_{g \in \mathcal{G}}(\text{CE}(g))^2 + \epsilon^2$, with probability $\geq 1 - \delta$.*

We will analyze each step of our algorithm and then combine the pieces to get Theorem 4.1. As we mention in the main text, step 3 is the main step, so Lemma D.2 is one of the core parts of our proof. Step 2 is where we construct a binning scheme so that each bin has an equal number of points—we show that this property holds approximately in the population (Lemma 4.3). This is important as well, particularly to ensure we can estimate the calibration error. Step 1 is basically Platt scaling, and the asymptotic analysis is fairly standard.

**Step 3**: Our proofs will require showing convergence in $\ell_2$ and $\ell_1$ norm in function space, which we define below:

**Definition D.1** (Distances between functions). *Given $f, g : [0,1] \to [0,1]$, for the $\ell_2$ norm we define $||f - g||_2^2 = \mathbb{E}[(f(Z) - g(Z))^2]$ and $||f - g||_2 = \sqrt{||f - g||_2^2}$. For the $\ell_1$ norm we define $||f - g||_1 = \mathbb{E}[|f(Z) - g(Z)|]$*

Recall that we showed that in the limit of infinite data the binned version of $g$, $g_\mathcal{B}$, has lower calibration error than $g$ (Proposition 3.3). However our method uses $n$ data points to empirically bin $g$, giving us $\hat{g}_\mathcal{B}$. We now show the key lemma that allows us to bound the calibration error and later the mean-squared error. That is, we show that the empirically binned function $\hat{g}_\mathcal{B}$ quickly converges to $g_\mathcal{B}$ in both $\ell_2$ and $\ell_1$ norms.

**Lemma D.2** (Empirical binning). *There exist constants $c_B, c_1, c_2$ such that the following is true. Given $g : [0,1] \to [0,1]$, binning set $T_3 = \{(z_i, y_i)\}_{i=1}^n$ and a 2-well-balanced binning scheme $\mathcal{B}$ of size $B$. Given $0 < \delta < 0.5$, suppose that $n \geq c_B B \log \frac{B}{\delta}$. Then with probability at least $1 - \delta$,*

$$||\hat{g}_\mathcal{B} - g_\mathcal{B}||_2 \leq \frac{c_2}{\sqrt{n}}\sqrt{\log \frac{B}{\delta}} \text{ and } ||\hat{g}_\mathcal{B} - g_\mathcal{B}||_1 \leq \frac{c_1}{\sqrt{nB}}\sqrt{\log \frac{B}{\delta}}$$

*Proof.* Recall that the intuition is in Figure 1 of the main text—the $g(z_i)$ values in each bin (gray circles in Figure 1c) are in a narrower range than the $y_i$ values (black crosses in Figure 1b) so when

we take the average we incur less of an estimation error. Now, there may be a small number of bins where the $g(z_i)$ values are not in a narrow range, but we will use the assumption that $\mathcal{B}$ is 2-well-balanced to show that these effects average out and the overall estimation error is small.

Define $R_j$ to be the set of $g(z_i)$ that fall into the $j$-th bin, given by $R_j = \{g(z_i) \mid g(z_i) \in I_j \wedge (z_i, y_i) \in T_3\}$ (recall that $T_3$ is the data we use in step 3). Let $p_j$ be the probability of landing in bin $j$, given by $p_j = \mathbb{P}(g(Z) \in I_j)$. Since $\mathcal{B}$ is 2-well-balanced, $p_j \geq \frac{1}{2B}$. Since $n \geq c_B B \log \frac{B}{\delta}$, by the multiplicative Chernoff bound, for some large enough $c_B$, with probability at least $1 - \frac{\delta}{2}$, $|R_j| \geq \frac{p_j}{2}$.

Consider each bin $j$. Let $\mu_j$ be the expected output of $g$ in bin $j$, given by $\mu_j = \mathbb{E}[g(Z) \mid g(Z) \in I_j]$. $\mu(R_j)$, the mean of the values in $R_j$, is the empirical average of $|R_j|$ such values, each bounded between $b_{j-1}$ and $b_j$ where $I_j = [b_{j-1}, b_j]$. So $\hat{\mu}(R_j)$ is sub-Gaussian with parameter:

$$\sigma^2 = \frac{(b_j - b_{j-1})^2}{4|R_j|} \leq \frac{(b_j - b_{j-1})^2}{2p_j n}$$

Then by the sub-Gaussian tail bound, for any $1 \leq j \leq B$, with probability at least $1 - \frac{\delta}{2B}$, we have:

$$(\mu_j - \hat{\mu}(R_j))^2 \leq \frac{(b_j - b_{j-1})^2}{p_j n} \log \frac{4B}{\delta} \tag{6}$$

So by union bound with probability at least $1 - \frac{\delta}{2}$ the above holds for all $1 \leq j \leq B$ simultaneously. We then bound the $\ell_2$-error.

$$
\begin{aligned}
\|\hat{g_\mathcal{B}} - g_\mathcal{B}\|_2 &= \sqrt{\sum_{j=1}^{B} p_j (\mu_j - \hat{\mu}(R_j))^2} \\
&\leq \sqrt{\sum_{j=1}^{B} p_j \frac{(b_j - b_{j-1})^2}{p_j n} \log \frac{4B}{\delta}} \qquad \text{(by equation (6))} \\
&= \sqrt{\frac{1}{n} \log \frac{4B}{\delta} \sum_{j=1}^{B} (b_j - b_{j-1})^2} \\
&\leq \sqrt{\frac{1}{n} \log \frac{4B}{\delta} \sum_{j=1}^{B} (b_j - b_{j-1})} \qquad \text{(because } 0 \leq b_j - b_{j-1} \leq 1) \\
&\leq \sqrt{\frac{1}{n} \log \frac{4B}{\delta}} \\
&\leq c_2 \frac{1}{\sqrt{n}} \sqrt{\log \frac{B}{\delta}}
\end{aligned}
$$

Similarly, we can also bound the $\ell_1$-error. Here we also use the fact that $p_j \leq \frac{2}{B}$ since $\mathcal{B}$ is 2-well-balanced.

$$\begin{aligned}
||\hat{g_\mathcal{B}} - g_\mathcal{B}||_1 &= \sum_{j=1}^{B} p_j |\mu_j - \hat{\mu}(R_j)| \\
&\leq \sum_{j=1}^{B} p_j \sqrt{\frac{(b_j - b_{j-1})^2}{p_j n} \log \frac{4B}{\delta}} \\
&= \sum_{j=1}^{B} \sqrt{\frac{p_j (b_j - b_{j-1})^2}{n} \log \frac{4B}{\delta}} \\
&\leq \sum_{j=1}^{B} \sqrt{\frac{2(b_j - b_{j-1})^2}{Bn} \log \frac{4B}{\delta}} \\
&\leq \sqrt{\frac{2}{Bn} \log \frac{4B}{\delta}} \sum_{j=1}^{B} (b_j - b_{j-1}) \\
&\leq c_1 \frac{1}{\sqrt{Bn}} \sqrt{\log \frac{B}{\delta}}
\end{aligned}$$

By union bound, these hold with probability at least $1 - \delta$, which completes the proof. $\qquad\square$

**Step 2**: Recall that we chose our bins so that each bin had an equal proportion of points in the recalibration set. In our proofs we required that this property (approximately) holds in the population as well. The following lemma shows this.

**Restatement of Lemma 4.3.** *For universal constant $c$, if $n \geq cB \log \frac{B}{\delta}$, with probability at least $1 - \delta$, the binning scheme $\mathcal{B}$ we chose is 2-well-balanced.*

*Proof.* Suppose we are given a bin construction set of size $n$, $T_n = \{(z_1, y_1), \ldots, (z_n, y_n)\}$. For any interval $I$, let $\hat{P}(I)$ be the empirical estimate of $P(I) = \mathbb{P}(g(Z) \in I)$ given by:

$$\hat{P}(I) = \frac{|\{(z_i, y_i) \in T_n \mid g(z_i) \in I\}|}{n}$$

We constructed the bins so that each interval $I_j$ contains $\frac{n}{B}$ points, or in other words, $\hat{P}(I_j) = \frac{1}{B}$. We want to show that $\frac{1}{2B} \leq \mathbb{P}(g(Z) \in I_j) \leq \frac{2}{B}$. Since the intervals are chosen from data, we want a uniform concentration result that holds for all such intervals $I_j$.

We will use a discretization argument. The idea is that we will cover $[0, 1]$ with $10B$ disjoint small intervals such that for each of these intervals $I'_j$, $P(g(Z) \in I'_j) = \frac{1}{10B}$. We will then use Bernstein and union bound to get that with probability at least $1 - \delta$, for all $I'_j$, $|P(I'_j) - \hat{P}_j(I'_j)| \leq \frac{1}{100B}$. Given an arbitrary interval $I$, we can write it as an approximate union of these small intervals, which will allow us to concentrate $|P(I) - \hat{P}(I)|$.

**Concentrating the small intervals:** Fix some interval $I'_j$. Let $w_i = \mathbb{I}(g(z_i) \in I'_j)$ for $i = 1, \ldots, n$. Then $w_i \sim \text{Bernoulli}(\frac{1}{10B})$. $\hat{P}(I'_j)$ is simply the empirical average of $n$ such values and as such with probability at least $1 - \frac{\delta}{10B}$:

$$|P(I'_j) - \hat{P}_j(I'_j)| \leq \sqrt{\frac{2}{10Bn} \log \frac{10B}{\delta}} + \frac{2}{3n} \log \frac{10B}{\delta}$$

If $n = cB \log \frac{B}{\delta}$ for a large enough constant $c$, we get:

$$|P(I'_j) - \hat{P}_j(I'_j)| \leq \frac{1}{100B}$$

And this was with probability at least $1 - \frac{\delta}{10B}$. So by union bound we get that with probability at least $1 - \delta$ this holds for all $I'_j$.

**Concentrating arbitrary intervals:** Now consider arbitrary $I \subseteq [0, 1]$. We can approximately write $I$ as a union of the small $I'_j$ intervals. More concretely, we can form a lower bound for $\hat{P}(I)$ by considering all $I'_j$ contained in $I$:

$$S_L = \{I'_j \mid I'_j \subseteq I\}$$

Similarly we can form an upper bound for $\hat{P}(I)$ by considering all $I'_j$ that have non-empty intersection with $I$:

$$S_U = \{I'_j \mid I'_j \cap I \neq \emptyset\}$$

We can then show:

$$\frac{9}{10}P(I) - \frac{1}{5B} \leq \hat{P}(I) \leq \frac{11}{10}P(I) + \frac{1}{5B}$$

Since in our case for all $j$, $\hat{P}(I_j) = \frac{1}{B}$, this gives us:

$$\frac{1}{2B} \leq P(I_j) \leq \frac{2}{B}$$

$\square$

**Step 1**: Recall that step 1 essentially applies a scaling method—we fit a small number of parameters to the recalibration data. We show that if $\mathcal{G}$ contains $g^* \in \mathcal{G}$ with low calibration error, then the empirical risk minimizer $g \in \mathcal{G}$ of the mean-squared loss will also quickly converge to low calibration error. Intuitively, methods like Platt scaling fit a single parameter to the data so standard results in asymptotic statistics tell us they will converge quickly to their optimal error, at least in mean-squared error. We can combine this with a decomposition of the mean-squared error into calibration and refinement, and the injectivity of $g \in \mathcal{G}$, to show they also converge quickly in calibration error.

**Lemma D.3** (Convergence of scaling). *Given $\delta$, there exists a constant $c$, such that for all $n$, $(\mathrm{CE}(g))^2 \leq \min_{g' \in \mathcal{G}} (\mathrm{CE}(g'))^2 + \frac{c}{n}$, with probability at least $1 - \delta$.*

*Proof.* **From calibration error to mean-squared error**: We use the classic decomposition of the mean-squared error into calibration error (also known as reliability) and refinement[2]. For any $g \in \mathcal{G}$ we have:

$$\mathrm{MSE}(g) = \underbrace{(\mathrm{CE}(g))^2}_{\text{calibration}} + \underbrace{\mathbb{E}[(\mathbb{E}[Y \mid g(Z)] - Y)^2]}_{\text{refinement}}$$

Note that the refinement term is constant for all injective $g \in \mathcal{G}$, since for injective $g$:

$$\mathbb{E}[(\mathbb{E}[Y \mid g(Z)] - Y)^2] = \mathbb{E}[(\mathbb{E}[Y \mid Z] - Y)^2]$$

This means that the difference in calibration error between any $g$ and $g'$ is precisely the difference in the mean-squared error. So it suffices to upper bound the generalization gap $\mathrm{MSE}(g) - \mathrm{MSE}(g^*)$ for the mean-squared error. Our analysis is fairly standard: we will show asymptotic convergence in the parameter space, and then use a Taylor expansion to show convergence in the MSE loss.

**Parameter convergence**: Recall that $\hat{\theta}$ denotes the parameters estimated by optimizing the empirical mean-squared error objective on $n$ samples in step 1 of our algorithm, and $\theta^*$ denotes the optimal parameters that minimize the mean-squared error objective on the population. From Theorem 5.23 of [48], on the asymptotic parameter convergence of M-estimators, we have as $n \to \infty$:

$$\sqrt{n}(\hat{\theta} - \theta^*) \to_D N(0, \Sigma)$$

Then for each $1 \leq i \leq d$, we have:

$$\sqrt{n}(\hat{\theta}_i - \theta_i^*) \to_D N(0, \sigma_i^2)$$

We will show that there exists $c_i$ such that for each $i$ and for all $n$, with probability at least $1 - \frac{\delta}{d}$:

$$|\hat{\theta}_i - \theta_i^*| \leq \frac{c_i}{n}$$

To see this, we begin with the definition of convergence in distribution, which says that the CDFs converge pointwise at every point where the CDF is continuous, which for a Gaussian is every point. That is, letting $z_i$ be a sample from $N(0, \sigma_i^2)$, we have for all $c$:

$$\lim_{n \to \infty} \mathbb{P}(\sqrt{n}(\hat{\theta}_i - \theta_i^*) \geq c) = \mathbb{P}(z_i \geq c)$$

By considering the CDF at each point and its negative, we can show the same result for the absolute value:

$$\lim_{n \to \infty} \mathbb{P}(\sqrt{n}|\hat{\theta}_i - \theta_i^*| \geq c) = \mathbb{P}(|z_i| \geq c)$$

The tails of a normal distribution are bounded, so we can choose $c_i'$ such that:

$$\mathbb{P}(|z_i| \geq c_i') \leq \frac{\delta}{2d}$$

By definition of limit, this means that we can choose $N_i$ such that for all $n \geq N_i$, we have:

$$\mathbb{P}(\sqrt{n}|\hat{\theta}_i - \theta_i^*| \geq c_i') \leq \frac{\delta}{d}$$

In other words, for all $n \geq N_i$, with probability at least $1 - \frac{\delta}{d}$:

$$|\hat{\theta}_i - \theta_i^*| \leq \frac{c_i'}{\sqrt{n}}$$

Since this only does not hold for finitely many values $1, \cdots, N_i - 1$, we can 'absorb' these cases into the constant. That is, for each $n \in \{1, \cdots, N_i - 1\}$, there exists $r_n$ such that if we use $n$ samples, then except with probability $\frac{\delta}{d}$, $|\hat{\theta}_i - \theta_i^*| \leq r_n$. So then we can choose $c_i$ such that for all $n$:

$$|\hat{\theta}_i - \theta_i^*| \leq \frac{c_i' + \max_{1 \leq m < N_i} r_m \sqrt{m}}{\sqrt{n}} \leq \frac{c_i}{\sqrt{n}}$$

We apply union bound over the indices $i$, and can then bound the $\ell_2$-norm of the difference between the estimated and optimal parameters, so that we can choose $k$ such that for all $n$, with probability at least $1 - \delta$:

$$||\hat{\theta} - \theta^*||_2^2 \leq \frac{k}{n}$$

**Loss convergence**: We denote the loss by $L$, defined as:

$$L(\theta) = \mathrm{MSE}(g_\theta) = \mathbb{E}[(Y - g_\theta(X))^2]$$

We approximate the loss $L$ by the first few terms of its Taylor expansion, which we denote by $\widetilde{L}$:

$$\widetilde{L}(\theta) = L(\theta^*) + \nabla L(\theta^*)^T (\hat{\theta} - \theta^*) + (\hat{\theta} - \theta^*)^T \nabla^2 L(\theta^*)(\hat{\theta} - \theta^*)$$

We assumed that $L$ was twice differentiable with continuous second derivative, and that $\theta^*$ minimized the loss in an open set, so $\nabla L(\theta^*) = 0$, and we also have (see e.g. Theorem 3.3.18 in [49]):

$$\lim_{||\hat{\theta} - \theta^*||_2 \to 0} \frac{L(\hat{\theta}) - \widetilde{L}(\hat{\theta})}{||\hat{\theta} - \theta^*||_2^2} = 0$$

By the definition of a limit if we fix $\epsilon > 0$, there exists $R > 0$ such that if $||\hat{\theta} - \theta^*||_2 \leq R$ then $L(\hat{\theta}) - \widetilde{L}(\hat{\theta}) \leq \epsilon ||\hat{\theta} - \theta^*||_2^2$. For some large enough $N_0$, if $n \geq N_0$, then with probability at least $1 - \delta$, $||\hat{\theta} - \theta^*||_2 \leq R$. As before, since this only does not hold for finitely many $N$, we can fold these cases into the constant so that there exists $\epsilon'$ such that for all $n$, $L(\hat{\theta}) - \widetilde{L}(\hat{\theta}) \leq \epsilon' ||\hat{\theta} - \theta^*||_2^2$ with probability at least $1 - \delta$. Plugging in $\widetilde{L}(\hat{\theta})$, we have:

$$L(\hat{\theta}) - L(\theta^*) \leq (\hat{\theta} - \theta^*)^T \nabla^2 L(\theta^*)(\hat{\theta} - \theta^*) + \epsilon' ||\hat{\theta} - \theta^*||_2^2$$

We can bound this by the operator norm of the Hessian, and then use the parameter convergence result:

$$L(\hat{\theta}) - L(\theta^*) \leq (||\nabla^2 L(\theta^*)||_{op} + \epsilon') ||\hat{\theta} - \theta^*||_2^2 \leq \frac{c}{n}$$

which holds with probability at least $1 - \delta$, as desired.

$\square$

Finally, we have the tools to prove the main theorem:

*Proof of Theorem 4.1.* The proof pieces together Lemmas D.3, D.2, 4.3 and Proposition 3.3.

For any fixed $c_1 > 0$, there exists $c_1'$ such that if $n \geq c_1'\left(\frac{1}{\epsilon^2}\right)$, from Lemma D.3, step 1 of our algorithm gives us $g$ with $(\mathrm{CE}(g))^2 \leq \min_{g' \in \mathcal{G}}(\mathrm{CE}(g'))^2 + c_1 \epsilon^2$, with probability at least $1 - \frac{\delta}{3}$.

Next, for universal constant $c_2$, if $n \geq c_2(B \log \frac{B}{\delta})$, from Lemma 4.3, step 2 chooses a 2-well-balanced binning scheme $\mathcal{B}$ with probability at least $1 - \frac{\delta}{3}$.

From Proposition 3.3, $(\mathrm{CE}(g_\mathcal{B}))^2 \leq (\mathrm{CE}(g))^2 \leq \min_{g' \in \mathcal{G}}(\mathrm{CE}(g'))^2 + c_1 \epsilon^2$. Then from Lemma D.2, for any $c_3 > 0$, there exists $c_3'$ such that if $n \geq c_3'\left(\frac{1}{\epsilon^2} \log \frac{B}{\delta}\right)$, step 3 gives us $\hat{g_\mathcal{B}}$ with $||\hat{g_\mathcal{B}} - g_\mathcal{B}||_2 \leq c_3 \epsilon$ with probability at least $1 - \frac{\delta}{3}$. We want to say that since $\hat{g_\mathcal{B}}$ is close to $g_\mathcal{B}$ and $g_\mathcal{B}$ has low calibration error, this must mean that $\hat{g_\mathcal{B}}$ has low calibration error.

To do this we represent the ($\ell_2$) calibration error of any $g$ as the distance between $g$ and a perfectly recalibrated version of $g$. That is, we define the perfectly recalibrated version of $g$ as:

$$\omega(g)(z) = \mathbb{E}[Y \mid g(Z) = z]$$

Then for any $g$, we can write $\mathrm{CE}(g) = ||g - \omega(g)||_2$. By triangle inequality on the $\ell_2$ norm on functions, we have:

$$||\hat{g_\mathcal{B}} - \omega(g_\mathcal{B})||_2 \leq ||\hat{g_\mathcal{B}} - g_\mathcal{B}||_2 + ||g_\mathcal{B} - \omega(g_\mathcal{B})||_2 \leq c_3 \epsilon + \sqrt{\min_{g' \in \mathcal{G}}(\mathrm{CE}(g'))^2 + c_1 \epsilon^2}$$

Now the LHS is not quite the calibration error of $\hat{g_\mathcal{B}}$, which is $||\hat{g_\mathcal{B}} - \omega(\hat{g_\mathcal{B}})||_2$ [3]. However, since $g$ is injective, $g_\mathcal{B}$ takes on a different value for each interval $I_j \in \mathcal{B}$. If $\hat{g_\mathcal{B}}$ also takes on a different value for each interval $I_j \in \mathcal{B}$, then we can see that $\omega(g_\mathcal{B}) = \omega(\hat{g_\mathcal{B}})$. If not, $\omega(\hat{g_\mathcal{B}})$ can only merge some of the intervals of $\omega(g_\mathcal{B})$, and by Jensen's we can show:

$$||\hat{g_\mathcal{B}} - \omega(\hat{g_\mathcal{B}})||_2 \leq ||\hat{g_\mathcal{B}} - \omega(g_\mathcal{B})||_2 \leq c_3 \epsilon + \sqrt{\min_{g' \in \mathcal{G}}(\mathrm{CE}(g'))^2 + c_1 \epsilon^2}$$

An alternative way to see this is to add infinitesimal noise to $\hat{g_\mathcal{B}}$ for each interval $I_j$, in which case we get $\omega(g_\mathcal{B}) = \omega(\hat{g_\mathcal{B}})$. Finally we convert back from CE to the squared calibration error:

$$(\mathrm{CE}(\hat{g_\mathcal{B}}))^2 = ||\hat{g_\mathcal{B}} - \omega(\hat{g_\mathcal{B}})||_2^2 \leq \min_{g' \in \mathcal{G}}(\mathrm{CE}(g'))^2 + (c_3^2 + c_1)\epsilon^2 + 2\sqrt{(c_3^2\epsilon^2)\left(\min_{g' \in \mathcal{G}}(\mathrm{CE}(g'))^2 + c_1 \epsilon^2\right)}$$

By the AM-GM inequality, we have:

$$2\sqrt{(c_3^2\epsilon^2)\left(\min_{g' \in \mathcal{G}}(\mathrm{CE}(g'))^2 + c_1 \epsilon^2\right)} \leq (c_3^2 + c_1)\epsilon^2 + \min_{g' \in \mathcal{G}}(\mathrm{CE}(g'))^2$$

Combining these, we get:

$$(\mathrm{CE}(\hat{g_\mathcal{B}}))^2 \leq 2 \min_{g' \in \mathcal{G}}(\mathrm{CE}(g'))^2 + 2(c_3^2 + c_1)\epsilon^2$$

By e.g. choosing $c_1 = 0.1$ and $c_3 = 0.1$, we have $2(c_3^2 + c_1) \leq 1$, which gives us the desired result. By union bound over each step, we have this with probability at least $1 - \delta$.

$\square$

## D.2 Bounding the mean-squared error

We also show that if we use lots of bins, discretization has little impact on model quality as measured by the mean-squared error. Note that recalibration itself typically *reduces/improves* the mean-squared error. However, in our method after fitting a recalibration function like Platt scaling does, we discretize the function outputs. This reduces the calibration error and allows us to measure the calibration error, but it does increase the mean-squared error by a small amount. Here we upper bound the increase in mean-squared error. In other words, our method allows for the calibration error of the final model to be measured, and has little impact on the mean-squared error.

**Proposition D.4** (MSE Bound). *If $\mathcal{B}$ is a 2-well-balanced binning scheme of size $B$ and $B = \widetilde{\Omega}(n)$, where $\widetilde{\Omega}$ hides $\log$ factors, then $\mathrm{MSE}(\hat{g}_{\mathcal{B}}) \leq \mathrm{MSE}(g) + O(\frac{1}{B})$.*

To show this we begin with a lemma showing that if $f$ and $g$ are close in $\ell_1$ norm, then their mean-squared errors are close:

**Lemma D.5.** *For $f, g : [0, 1] \to [0, 1]$, $\mathrm{MSE}(f) \leq \mathrm{MSE}(g) + 2||f - g||_1$.*

*Proof.*

$$
\begin{aligned}
\mathbb{E}[(f(Z) - Y)^2 - (g(Z) - Y)^2] &= \mathbb{E}[(f(Z) - g(Z))(f(Z) + g(Z) - 2Y)] \\
&\leq \mathbb{E}[|f(Z) - g(Z)||f(Z) + g(Z) - 2Y|] \\
&\leq \mathbb{E}[2|f(Z) - g(Z)|] \\
&= 2||f - g||_1
\end{aligned}
$$

Where the third line followed because $-2 \leq f(Z) + g(Z) - 2Y \leq 2$. $\qquad\square$

Next, we show that in the limit of infinite data, if we bin with a well-balanced binning scheme then the MSE cannot increase by much.

**Lemma D.6.** *Let $\mathcal{B}$ be an $\alpha$-well-balanced binning scheme of size $B$. Then $\mathrm{MSE}(g_{\mathcal{B}}) \leq \mathrm{MSE}(g) + \frac{2\alpha}{B}$.*

*Proof.* We bound $||g_{\mathcal{B}} - g||_1$ and then use Lemma D.5. We use the law of total expectation, conditioning on $\beta(g(Z))$, the bin that $g(Z)$ falls into.

$$
\begin{aligned}
||g_{\mathcal{B}} - g||_1 &= \mathbb{E}[|g_{\mathcal{B}}(Z) - g(Z)|] \\
&\leq \mathop{\mathbb{E}}_{\beta(g(Z))} \left[ \mathop{\mathbb{E}}_{Z|\beta(g(Z))} [|g_{\mathcal{B}}(Z) - g(Z)|] \right] \\
&\leq \mathop{\mathbb{E}}_{\beta(g(Z))} \left[ b_{\beta(g(Z))} - b_{\beta(g(Z))-1} \right]
\end{aligned}
$$

We now use the fact that $\mathcal{B}$ is $\alpha$-well-balanced.

$$
\begin{aligned}
\mathop{\mathbb{E}}_{\beta(g(Z))} \left[ (b_{\beta(g(Z))} - b_{\beta(g(Z))-1}) \right] &= \sum_{i=1}^{B} \mathbb{P}\big(g(Z) \in [b_{\beta(g(Z))-1}, b_{\beta(g(Z))}]\big)(b_{\beta(g(Z))} - b_{\beta(g(Z))-1}) \\
&\leq \sum_{i=1}^{B} \frac{\alpha}{B}(b_{\beta(g(Z))} - b_{\beta(g(Z))-1}) \\
&\leq \frac{\alpha}{B}
\end{aligned}
$$

Finally, from Lemma D.5, we get that $\mathrm{MSE}(g_{\mathcal{B}}) \leq \mathrm{MSE}(g) + \frac{2\alpha}{B}$. $\qquad\square$

The above lemma bounds the increase in MSE due to binning in the infinite sample case – next we deal with the finite sample case and prove proposition D.4:

*Proof of Proposition D.4:* Ignoring all $\log$ factors, from Theorem D.2 if $n = \widetilde{\Omega}(B)$, we have $||\hat{g}_{\mathcal{B}} - g_{\mathcal{B}}||_1 = O(\frac{1}{\sqrt{nB}})$. Then from Lemma D.5, $\mathrm{MSE}(\hat{g}_{\mathcal{B}}) \leq \mathrm{MSE}(g_{\mathcal{B}}) + O(\frac{1}{\sqrt{Bn}}) \leq \mathrm{MSE}(g_{\mathcal{B}}) + O(\frac{1}{B})$. From Theorem D.6, since $\mathcal{B}$ is 2-well-balanced, we have $\mathrm{MSE}(g_{\mathcal{B}}) \leq \mathrm{MSE}(g) + O(\frac{1}{B})$. This gives us $\mathrm{MSE}(\hat{g}_{\mathcal{B}}) \leq \mathrm{MSE}(g) + O(\frac{1}{B})$. $\qquad\square$

### D.3 Alternative binning schemes

We note that there are alternative binning schemes in the literature. For example, the $B$ bins can be chosen as $I_1 = [0, \frac{1}{B}], I_2 = (\frac{1}{B}, \frac{2}{B}], \ldots, I_B = (\frac{B-1}{B}, 1]$. The main problem with this binning scheme is that we may not be able to measure the calibration error efficiently, which is critical. However, if we choose the bins like this, and are lucky that the binning scheme happens to be 2-well-balanced, we can improve the bounds on the MSE that we proved above. This motivates

(a) Effect of number of bins $B$ on top calibration error on ImageNet.

(b) Effect of number of bins $B$ on top calibration error on CIFAR-10.

Figure 7: Recalibrating using 1,000 data points on ImageNet and CIFAR-10, the scaling-binning calibrator typically achieves lower squared calibration error than histogram binning, especially when the number of bins $B$ is large. The difference is very significant on ImageNet, where our method does better when $B \geq 10$, and gets a nearly 5 times lower calibration error when $B = 100$. For CIFAR-10 our method does better when $B > 30$, which supports the theory, which predicts that our method does better when $B$ is large. However, when $B$ is small, practitioners should try both histogram binning and the scaling-binning calibrator.

alternative hybrid binning schemes, where we try to keep the width of the bins as close to $1/B$ as possible, while ensuring that each bin contains lots of points as well. We think analyzing what binning schemes lead to the best bounds, and seeing if this can improve the calibration method, is a good direction for future research.

# E    Experimental details and ablations for section 4

We give more experimental details for our CIFAR-10 experiment, show experimental results for top-label calibration in ImageNet and CIFAR-10, and give details and results for our synthetic experiments. Note that the code is available in the supplementary folder for completeness.

**Experimental details**: We detail our experimental protocol for CIFAR-10 first. The CIFAR-10 validation set has 10,000 data points. We sampled, with replacement, a recalibration set of 1,000 points. In our theoretical approach and analysis, we split up these sets into multiple parts. For example, we used the first part for training a function, second part for bin construction, third part for binning. In practice, using the same set for all three steps worked out better, for both histogram binning and the scaling-binning calibrator. We believe that there may be theoretical justification for merging these sets, although we leave that for future work. For the marginal calibration experiment we ran either the scaling-binning calibrator (we fit a sigmoid in the function fitting step) or histogram binning. We calibrated each of the $K$ classes seprately as described in Section 2, and measured the marginal calibration error on the entire set of 10K points. We repeated this entire procedure 100 times, and computed mean and 90% confidence intervals.

In this experiment, we are checking a very precise hypothesis—assuming that the empirical distribution on the 10,000 validation points is the true data distribution, how do these methods perform? This is similar to the experimental protocol used in e.g. [20]. An alternative experimental protocol would have been to first split the CIFAR-10 data into two sets of size $(1000, 9000)$. We could have then used the first set to recalibrate the model using either the scaling-binning calibrator or histogram binning, and then used the remaining 9,000 examples to estimate the calibration error on the ground truth distribution, using Bootstrap to compute confidence intervals. However, when we ran this experiment, we noticed that the results were very sensitive to which set of 1,000 points we used to recalibrate. Multiple runs of this experiment led to very different results. The point is that there are two sources of randomness—the randomness in the data the recalibration method operates on, and the randomness in the data used to evaluate and compare the recalibrators. In our protocol we account for both of these sources of randomness.

**Top-label calibration experiments**: We also ran experiments on top-label calibration, for both ImageNet and CIFAR-10. The protocol is exactly as described above, except instead of calibrating each of the $K$ classes, we calibrated the top probability prediction of the model. More concretely, for each input $x_i$, the uncalibrated model outputs a probability $p_i$ corresponding to its top prediction $k_i$, where the true label is $y_i$. We create a new dataset $\{(p_1, \mathbb{I}(k_1 = y_1)), \ldots, (p_n, \mathbb{I}(k_n = y_n))\}$ and run the scaling-binning calibrator (fitting a sigmoid in the function fitting step, as in Platt scaling) or histogram binning on this dataset, using $B$ bins. This calibrates the probability corresponding to the top prediction of the model. We evaluate the recalibrated models on the top-label calibration error metric described in Section 2. For both CIFAR-10 and ImageNet we sampled, with replacement, a recalibration set of 1,000 points for the recalibration data, and we measured the calibration error on the entire set (10,000 points for CIFAR-10, and 50,000 points for ImageNet) as above. We show $90\%$ confidence intervals for all plots.

Figure 7a shows that on ImageNet the scaling-binning calibrator gets significantly lower calibration errors than histogram binning when $B \geq 10$, and nearly a 5 times lower calibration error when $B = 100$. Both methods get similar calibration errors when $B = 1$ or $B = 5$. Figure 7b shows that on CIFAR-10 when $B$ is high, the scaling-binning calibrator gets lower calibration errors than histogram binning, but when $B$ is low histogram binning gets lower calibration errors. We believe that the difference might be because the CIFAR-10 model is highly accurate at top-label prediction to begin with, getting an accuracy of over $93\%$, so there is not much scope for re-calibration. In any case, this ablation tells us that practitioners should try multiple methods when recalibrating their models and evaluate their calibration error.

**(A) Synthetic experiments to validate bounds**: We first describe the scaling family we use, which is Platt scaling after applying a log-transform [12], otherwise known as beta calibration [50]. Let $\sigma$ be the standard sigmoid function given by:

$$\sigma(x) = \frac{1}{1 + \exp(-x)}$$

Then, our recalibration family $\mathcal{G}$ consists of $g$ parameterized by $a, c$, given by:

$$g(z; a, c) = \sigma\left(a \log \frac{z}{1 - z} + c\right)$$

In this set of synthetic experiments, we assume well-specification, that is $P(Y = 1 \mid Z = z) = g(z; a, c)$ for some $a, c$. We set $P(Z) = \text{Uniform}[0, 1]$. Since we know $P(Y = 1 \mid Z)$, we can approximate the true squared calibration error in this case, even for scaling methods. To do this, we sample $m = 10000$ points $z_1, \ldots, z_m$ independently from $P(Z)$. An *unbiased* estimate of the squared calibration error then is:

$$(\text{CE}(g))^2 \approx \frac{1}{m} \sum_{i=1}^{m} \left[P(Y \mid Z = z_i) - g(z_i)\right]^2$$

For each $n$ (number of recalibration samples) and $B$ (number of bins), we run either histogram binning or the scaling-binning calibrator with scaling family $\mathcal{G}$ and evaluate its calibration error as described above. We repeat this 1000 times, and compute $90\%$ confidence intervals. We fix $a = 2$ and $c = 1$.

In the first sub-experiment we fix $B = 10$ and vary $n$, plotting $1/\epsilon^2$ in Figure 8 (recall that $\epsilon^2$ is the squared calibration error). We plot the calibration errors for each method in a different plot because of the difference in scales, the scaling-binning calibrator achieves a much lower calibration error than histogram binning. As the theory predicts, we see that $1/\epsilon^2$ is approximately linear in $n$ for both calibrators. For example, when $B = 10$ if we increase from $n = 1000$ to $n = 2000$ the squared calibration error of histogram binning decreases by $2.00 \pm 0.06$ times, and the squared calibration error of our method decreases by $1.98 \pm 0.09$ times.

In the second sub-experiment we fix $n = 2000$ and vary $B$, plotting $1/\epsilon^2$ in Figure 11. For the scaling-binning calibrator $1/\epsilon^2$ is nearly constant (within the margin of error), but for histogram binning $1/\epsilon^2$ scales close to $1/B$. When $n = 2000$ and we increase from 5 to 20 bins, our method's squared calibration error decreases by $2\% \pm 7\%$ but for histogram binning it increases by $3.71 \pm 0.15$ *times*. For reference, we plot $P(Y \mid Z = z)$ in Figure 10a.

**(B) Synthetic experiments to compare the scaling-binning calibrator and the scaling method**: We run an illustrative toy experiment to show that there are some cases where the scaling-binning

(a) Histogram binning.

(b) The scaling-binning calibrator.

Figure 8: Plots of $1/\epsilon^2$ against $n$ (recall that $\epsilon^2$ is the squared calibration error). We see that for both methods $1/\epsilon^2$ increases approximately linearly with $n$, which match the theoretical bounds.

(a) Histogram binning.

(b) The scaling-binning calibrator.

Figure 9: Plots of $1/\epsilon^2$ against $b$ (recall that $\epsilon^2$ is the squared calibration error). Note that the $Y$ axis for the scaling-binning calibrator is clipped to 6600 and 7800 to show the relevant region. We see that for histogram binning $1/\epsilon^2$ scales close to $1/B$, in other words the calibration error increases with the number of bins (important note: the plot decreases because we plot the inverse $1/\epsilon^2$). For the scaling-binning calibrator $1/\epsilon^2$ is relatively constant, within the margin of estimation error, as predicted by the theory.

(a) $P(Y \mid Z = z)$ for Experiment (A)

(b) $P(Y \mid Z = z)$ for Experiment (B)

Figure 10: Plots of $P(Y \mid Z = z)$ against $z$ for both synthetic experiments.

Figure 11: Plot of $\epsilon^2$ (squared calibration error) against number of samples $n$ used to recalibrate. We can see in this case the scaling-binning calibrator consistently gets lower calibration error.

calibrator does better than the underlying scaling method—there are other cases where the underlying scaling method does better. the scaling-binning calibrator can do better because if we have infinite data, Proposition 3.3 showed that the binned version $g_{\mathcal{B}}$ has lower calibration error than $g$. On the other hand, in step 3 of the scaling-binning calibrator algorithm we empirically bin the outputs of the scaling method which incurs an estimation error, and could mean the scaling-binning calibrator has higher calibration error than the underlying scaling method. Our key advantage is that unlike scaling methods our method has measurable calibration error so if we are not calibrated we can get more data or use a different scaling family.

Building on the previous synthetic experiments, in this experiment, we set the ground truth $P(Y = 1 \mid Z = z) = g(z; a, c) + h(z)$ where for each $z$, $h(z) \in \{-0.02, 0.02\}$ with equal probability. In this case we set $P(Z) = \text{Uniform}[0.25, 0.75]$ so that $P(Y = 1 \mid Z = z) \in [0, 1]$. We fix $B = 10$ and vary $n$, plotting the squared calibration error $\epsilon^2$ in Figure 11. With $B = 10$ bins, $n = 3000$ the squared calibration error is $5.2 \pm 1.1$ times lower for the scaling-binning calibrator than the underlying scaling method using a sigmoid recalibrator. For reference, we plot $P(Y \mid Z = z)$ in Figure 10b.

# F   Proofs for section 5

In this section we prove the finite sample bounds for the plugin and debiased estimators. We follow a very similar structure for both the plugin estimator and the debiased estimators.

We first give a proof for the plugin estimator. At a high level, we decompose the plugin estimator into three terms (Lemma F.2), and then bound each of these terms. Most of these terms simply involve algebraic manipulation and standard concentration results, except Lemma F.4 which requires some tricky conditioning.

The debiased estimator decomposes into three terms as well, two of these terms are the same as those in the plugin estimator. Bounding the third term (Lemma F.8) is the key to the improved sample complexity of the plugin estimator. The debiased estimator is not completely unbiased. However, with high probability if we condition on the $x_i$s in the evaluation set, each of these error terms is unbiased. We can then use Hoeffding's to concentrate each term near 0. The errors in each bin are then independent which leads to some cancelations of the error terms when we sum them up.

**We use the following notation simplification** to simplify the theorem statements and proofs:

$$p_i = P(f(X) = s_i)$$
$$y_i^* = \mathbb{E}[Y \mid f(X) = s_i]$$
$$e_i = (s_i - y_i^*)$$

Then, if we let $E^{*2}$ denote the actual squared calibration error, we have:

$$E^{*2} = \sum_{i=1}^{b} p_i e_i^2$$

We begin by noting that $\hat{p}_i$ is close to $p_i$ for all $i$. This is a standard application of either Bernstein's inequality or the multiplicative Chernoff bound.

**Lemma F.1.** *Suppose $p_i > \frac{12}{n} \log \frac{2B}{\delta}$ for all $i$. Then we can define $c(n) < 0.5$ such that except with probability $\delta$ for all $i$ we have:*

$$|\hat{p}_i - p_i| < c(n)p_i := \sqrt{\frac{3}{n \min p_i} \log \frac{2B}{\delta}} p_i$$

## F.1   Analysis of plugin estimator (proof of Theorem 5.3)

The following lemma is crucial – we decompose the plugin estimator into three terms that we can bound separately.

**Lemma F.2** (Plugin decomposition). *The plugin estimator satisfies the following decomposition:*

$$\hat{\mathcal{E}}_{\text{pl}}^2 = \underbrace{\sum_{i=1}^{b} \hat{p}_i e_i^2}_{(P1)} - \underbrace{2 \sum_{i=1}^{b} \hat{p}_i e_i (\hat{y}_i - y_i^*)}_{(P2)} + \underbrace{\sum_{i=1}^{b} \hat{p}_i (\hat{y}_i - y_i^*)^2}_{(P3)}$$

*Proof.* The proof is by algebra:

$$\hat{\mathcal{E}}_{\text{pl}}^2 = \sum_{i=1}^{b} \hat{p}_i (s_i - \hat{y}_i)^2$$

$$= \sum_{i=1}^{b} \hat{p}_i [e_i - (\hat{y}_i - y_i^*)]^2$$

$$= \underbrace{\sum_{i=1}^{b} \hat{p}_i e_i^2}_{(P1)} - \underbrace{2 \sum_{i=1}^{b} \hat{p}_i e_i (\hat{y}_i - y_i^*)}_{(P2)} + \underbrace{\sum_{i=1}^{b} \hat{p}_i (\hat{y}_i - y_i^*)^2}_{(P3)}$$

$\square$

We now bound each of these three terms with the following three lemmas. We condition on $|\hat{p}_i - p_i| < c(n)p_i < 0.5p_i$ for all $i$, which holds with high probability from Lemma F.1.

**Lemma F.3.** *Let* $(P1)$ *be as defined in Lemma F.2. Suppose* $|\hat{p}_i - p_i| < c(n)p_i$ *for all* $i$. *Then*

$$|(P1) - E^{*2}| \leq c(n)E^{*2}$$

*Proof.* The proof is by algebra.

$$|(P1) - E^{*2}| = |\sum_{i=1}^{b} \hat{p}_i e_i^2 - \sum_{i=1}^{b} p_i e_i^2|$$

$$= |\sum_{i=1}^{b} (\hat{p}_i - p_i)e_i^2|$$

$$\leq \sum_{i=1}^{b} |(\hat{p}_i - p_i)|e_i^2$$

$$\leq \sum_{i=1}^{b} c(n)p_i e_i^2$$

$$\leq c(n) \sum_{i=1}^{b} p_i e_i^2$$

$$\leq c(n)E^{*2}$$

□

**Lemma F.4.** *Let* $(P2)$ *be as defined in Lemma F.2. Suppose* $|\hat{p}_i - p_i| < c(n)p_i < 0.5p_i$ *for all* $i$. *Then with probability* $\geq 1 - \delta$:

$$|(P2)| \leq \sqrt{\frac{2(1+c(n))E^{*2}}{n} \log \frac{2}{\delta}}$$

*Proof.* Recall that we evaluated our estimators on an independent and identically distributed evaluation set $T_n = \{(x_1, y_1), \ldots, (x_n, y_n)\}$. Also, note that since $|\hat{p}_i - p_i| < p_i$, $\hat{p}_i > 0$. Let $Z = (f(x_1), \cdots, f(x_n))$ be a random variable.

$\hat{y}_i$ simply takes the empirical average of the label values, and is therefore an unbiased estimator of $y_i^*$ even if we condition on $Z$:

$$\mathbb{E}[\hat{y}_i - y_i^* \mid Z] = 0$$

Next we look at the distribution of $\hat{y}_i - y_i^* \mid Z$. For all $(x_j, y_j) \in T_n$, $y_j \in \{0, 1\}$. Additionally, $\{y_j \mid (x_j, y_j) \in T_n\} \mid Z$ is also independently (but not identically) distributed. So by Hoeffding's lemma, $\hat{y}_i - y_i^* \mid Z$ is sub-Gaussian with parameter $\frac{1}{4\hat{p}_i n}$.

Here, we note that $\hat{p}_i$ is a constant given $Z$. Then, we get that $\hat{p}_i e_i (\hat{y}_i - y_i^*) \mid Z$ has expected value 0 and is sub-Gaussian with parameter:

$$\sigma_i^2 = \hat{p}_i^2 e_i^2 \frac{1}{4\hat{p}_i n} = \frac{\hat{p}_i e_i^2}{4n}$$

This means that the sum, $(P2)$ has expected value 0 and is sub-Gaussian with parameter:

$$\sigma^2 = 2^2 \sum_{i=1}^{B} \sigma_i^2 = 4 \sum_{i=1}^{B} \frac{\hat{p}_i e_i^2}{4n} \leq \frac{(1+c(n))E^{*2}}{n}$$

By applying the sub-Gaussian tail inequality, we get that with probability at least $1 - \delta$,

$$|(P2)| \leq \sqrt{\frac{2(1+c(n))E^{*2}}{n} \log \frac{2}{\delta}}$$

Since this was true for all $Z$, this is true if we marginalize over $Z$ as well, which completes the proof. □

**Lemma F.5.** *Let* $(P3)$ *be as defined in Lemma F.2. Suppose* $|\hat{p}_i - p_i| < c(n)p_i < 0.5p_i$ *for all* $i$. *Then with probability* $\geq 1 - \delta$:

$$|(P3)| \leq \frac{B}{2n} \log \frac{2B}{\delta}$$

*Proof.* Fix arbitrary $\hat{p}_i$s satisfying $|\hat{p}_i - p_i| < c(n)p_i < 0.5p_i$. Note that this gives us $\hat{p}_i > 0$.

By Hoeffding's bound, for any fixed $i$, with probability at least $1 - \frac{\delta}{b}$:

$$|\hat{y}_i - y_i^*| \leq \sqrt{\frac{1}{2\hat{p}_i n} \log \frac{2B}{\delta}}$$

Applying union bound over $i = 1, \cdots, B$, we get that the above holds *for all* $i$ with probability at least $1 - \delta$. Then with probability at least $1 - \delta$, for all $i$:

$$|\hat{p}_i(\hat{y}_i - y_i^*)^2| \leq \frac{1}{2n} \log \frac{2B}{\delta}$$

Summing over the bins $i = 1, \cdots, B$, we get:

$$0 \leq (P3) \leq \frac{B}{2n} \log \frac{2B}{\delta}$$

$\square$

Bounding the error of the plugin estimator simply involves combining the bounds for each of the terms, $P1$, $P2$, $P3$.

**Theorem F.6.** *Let* $p_i = P(f(X) = s_i)$ *and suppose* $p_i > \frac{12}{n} \log \frac{2B}{\delta}$ *for all* $i$. *Let* $c(n)$ *be defined as:*

$$c(n) = \sqrt{\frac{3}{n \min p_i} \log \frac{2B}{\delta}}$$

*Then for the plugin estimator, with probability at least* $1 - 3\delta$,

$$|\hat{E}_{pl}^2 - E^{*2}| \leq c(n)E^{*2} + \sqrt{\frac{2(1 + c(n))E^{*2}}{n} \log \frac{2}{\delta}} + \frac{B}{2n} \log \frac{2B}{\delta}$$

*Proof.* We have:

$$|\hat{\mathcal{E}}_{pl}^2 - E^{*2}| \leq |P1 - E^{*2}| + |P2| + |P3|$$

From Lemma F.1 we have $|\hat{p}_i - p_i| < c(n)p_i < 0.5p_i$ with probability $\geq 1 - \delta$. Conditioning on this, we combine Lemmas F.3, F.4, F.5 with union bound to get the desired result. $\square$

We then prove the final bound for the plugin estimator, which we recall below.

**Restatement of Theorem 5.3.** *Suppose we have a binned model with squared calibration error* $\mathcal{E}^2$, *where the binning scheme is 2-well-balanced, that is for all* $s \in S$, $\mathbb{P}(f(X) = s) \geq \frac{1}{2B}$. [4] *If* $n \geq c\frac{B}{\mathcal{E}^2} \log \frac{B}{\delta}$ *for some universal constant* $c$, *then for the plugin estimator, we have* $\frac{1}{2}\mathcal{E}^2 \leq \hat{\mathcal{E}}_{pl}^2 \leq \frac{3}{2}\mathcal{E}^2$ *with probability at least* $1 - \delta$.

*Proof.* This is now a simple corollary of Theorem F.6. For large enough constant $c$, where $c$ is a constant independent of all the other variables, we choose $n = c\frac{B}{\epsilon^2} \log \frac{B}{\epsilon^2}$. Plugging it into the bound of Theorem F.6, we get the desired result, that $|\hat{E}^2 - E^{*2}| \leq \frac{1}{2}E^{*2}$. Notice that the dominating term is term $(P3)$ in Theorem F.6—we will see that the debiased estimator improves on this.

In fact, we can also show that in the worst case the plugin estimator will need at least $O(\frac{B}{\epsilon^2})$ samples to estimate the calibration error. To see this, first note that the bias of the plugin estimator, which comes from term $(P3)$ is at least $\frac{B}{n}$. Furthermore, in the analysis of the debiased estimator we show that the variance of this term is on the order of $O(\frac{\sqrt{B}}{n})$. So if $n < 0.1\frac{B}{\epsilon^2}$ we can consider very large $B$, and use Chebyshev to show that that with high probability the estimation error is larger than $\epsilon^2$.

$\square$

## F.2 Analysis of debiased estimator (proof of Theorem 5.4)

Next, we bound the error of the debiased estimator. The proof follows along the lines of the plugin estimator. We begin with a decomposition (Lemma F.7), similar to the decomposition of the plugin estimator. However, one of the terms in the decomposition, $C3$, is different. Lemma F.8 bounds this term $C3$. The rest of the proof is the same as for the plugin estimator, so we omit the other proofs.

As with the plugin estimator, we have a decomposition for the debiased estimator.

**Lemma F.7** (Debiased decomposition). *The debiased estimator satisfies the following decomposition:*

$$\hat{\mathcal{E}}_{\text{db}}^2 = \underbrace{\sum_{i=1}^{b} \hat{p}_i e_i^2}_{(C1)} - \underbrace{2 \sum_{i=1}^{b} \hat{p}_i e_i (\hat{y}_i - y_i^*)}_{(C2)} + \underbrace{\sum_{i=1}^{b} \hat{p}_i \left[ (\hat{y}_i - y_i^*)^2 - \frac{\hat{y}_i(1 - \hat{y}_i)}{\hat{p}_i n - 1} \right]}_{(C3)}$$

As with the plugin estimator, we bound each of the three terms. Notice that $C1$ and $C2$ are the same as terms $P1$ and $P2$ in the plugin estimator decomposition, so the bounds for those carry over. The next lemma bounds the error in $C3$.

**Lemma F.8.** *Let* $(C3)$ *be as defined in Lemma F.7. Suppose* $|\hat{p}_i - p_i| < c(n)p_i < 0.5p_i$ *for all* $i$. *Then with probability* $\geq 1 - \delta$:

$$|(C3)| \leq \frac{3\sqrt{B}}{n} \log \frac{n}{\delta} + \frac{\delta}{n}$$

*Proof.* Let $Z = (f(x_1), \cdots, f(x_n))$ be a random variable. We note that for all $i$, $\hat{p}_i$ is a deterministic function of $Z$. For convenience, define $t_i$ as follows:

$$t_i = (\hat{y}_i - y_i^*)^2 - \frac{\hat{y}_i(1 - \hat{y}_i)}{\hat{p}_i n - 1}$$

**Computing the expectation:** The debiased estimator debiases the plugin estimator. In particular, we briefly explain why $\mathbb{E}[C3 \mid Z] = 0$. Since $\hat{y}_i$ is the mean of $n\hat{p}_i$ draws of a Bernoulli with parameter $y_i^*$, we have:

$$\mathbb{E}[(\hat{y}_i - y_i^*)^2 \mid Z] = \frac{y_i^*(1 - y_i^*)}{n\hat{p}_i}$$

The term we subtracted is the unbiased estimate of the standard deviation of the samples, so from elementary statistics:

$$\mathbb{E}\left[ \frac{\hat{y}_i(1 - \hat{y}_i)}{\hat{p}_i n - 1} \mid Z \right] = \frac{y_i^*(1 - y_i^*)}{n\hat{p}_i}$$

Which implies that $\mathbb{E}[C3 \mid Z] = 0$.

**Bounding each term:** By Hoeffding's bound, for any fixed $i$, we get that with probability at least $1 - \frac{\delta}{n}$:

$$|\hat{y}_i - y_i^*| \leq \sqrt{\frac{1}{2\hat{p}_i n} \log \frac{2n}{\delta}}$$

Let $E_i$ be the event that this is indeed the case. Condition on $E_i$ holding for all $i$ – by union bound this happens with probability at least $1 - \delta$. With some algebra, we then get:

$$|\hat{p}_i t_i| = \left| \hat{p}_i \left[ (\hat{y}_i - y_i^*)^2 - \frac{\hat{y}_i(1 - \hat{y}_i)}{\hat{p}_i n - 1} \right] \right| \leq \frac{3}{2n} \log \frac{B}{\delta}$$

**Concentration:** Next, we analyze the concentration of $T = \left[ (C3) \mid Z, \forall i.E_i \right]$ around its mean $\mu$. $|\hat{p}_i t_i|$ is bounded so is sub-Gaussian with parameter:

$$\sigma_i^2 = \frac{9}{4n^2} \log \frac{B}{\delta}$$

Each term $\hat{p}_i t_i$ in the sum is independent, even when conditioned on $Z$. So $T$ is sub-Gaussian with parameter:

$$\sigma^2 = \sum_{i=1}^{B} \sigma_i^2 = \frac{9B}{4n^2} \log \frac{B}{\delta}$$

So by the sub-Gaussian tail bound, we have:

$$|T - \mu| \leq \sqrt{2\sigma^2 \log \frac{1}{\delta}} \leq \frac{3\sqrt{2}}{2} \frac{\sqrt{B}}{n} \sqrt{\log \frac{n}{\delta} \log \frac{1}{\delta}}$$

This can be simplified to:

$$|T - \mu| \leq \frac{3\sqrt{B}}{n} \log \frac{n}{\delta}$$

**Bounding the bias:** Although $\mathbb{E}[C3 \mid Z] = 0$, conditioning on $E_i$ introduces some bias. However, we can show this bias is small. First, notice that $|t_i| \leq 1$. The event $E_i$ holds with probability at least $1 - \frac{\delta}{n}$. Then by the law of total expectation, conditioning on $E_i$ shifts the mean by at most $\frac{\delta}{n}$ – in other words $|\mathbb{E}[t_i \mid E_i, Z]| \leq \frac{\delta}{n}$. Summing over $t_i$s, we get:

$$|\mathbb{E}[(C3) \mid Z, \forall i.E_i]| \leq \sum_{i=1}^{B} \hat{p}_i |\mathbb{E}[t_i \mid E_i, Z]| \leq \frac{\delta}{n}$$

**Finishing up:** Combining the bias and concentration, we get that with probability at least $1 - 2\delta$:

$$|(C3)| \leq \frac{3\sqrt{B}}{n} \log \frac{n}{\delta} + \frac{\delta}{n}$$

$\square$

We combine the bounds for $(C1)$, $(C2)$, $(C3)$, as in Theorem F.6, to bound the estimation error of the debiased estimator.

**Theorem F.9.** *In the same setting as Theorem F.6, for the debiased estimator, with probability at least $1 - 4\delta$,*

$$|\hat{E}^2 - E^{*2}| \leq c(n)E^{*2} + \sqrt{\frac{2(1 + c(n))E^{*2}}{n} \log \frac{2}{\delta}} + \frac{3\sqrt{B}}{n} \log \frac{n}{\delta} + \frac{\delta}{n}$$

We interpret the bound in Theorem F.9 in two regimes. In the first regime, we fix the problem parameters $p_i, E^{*2}$, and look at what happens as we send $n$ to infinity. In that case, the second term dominates, and we see that the error is approximately proportional to $\frac{1}{\sqrt{n}}$, which is the same as for the plugin estimator. However, in general we do not need to estimate the calibration error extremely finely, and may be satisfied as long as we estimate the calibration error within a constant multiplicative factor. That is, we might only need $n$ to be large enough so that our estimate $\hat{E}^2$ is on the right order, e.g. between $0.5E^{*2}$ and $1.5E^{*2}$ (where $0.5$ and $1.5$ can be replaced by other constants). In that regime, the third term dominates and the error is approximately proportional to $\frac{\sqrt{B}}{n}$, which is better than for the plugin estimator where it is proportional to $\frac{B}{n}$ (see Theorem F.6). This is captured in the final bound, where the proof closely parallels that of Theorem 5.3.

**Restatement of Theorem 5.4.** *Suppose we have a binned model with squared calibration error $\mathcal{E}^2$ and for all $s \in S$, $\mathbb{P}(f(X) = s) \geq \frac{1}{2B}$. If $n \geq c\frac{\sqrt{B}}{\mathcal{E}^2} \log \frac{B}{\delta}$ for some universal constant $c$ then for the debiased estimator, we have $\frac{1}{2}\mathcal{E}^2 \leq \hat{\mathcal{E}}_{\text{db}}^2 \leq \frac{3}{2}\mathcal{E}^2$ with probability at least $1 - \delta$.*

# G Additional experiments for section 5

## G.1 Debiasing the ECE

We propose a way to more accurately estimate the $\ell_1$ calibration error (popularly known as ECE), and run experiments on ImageNet and CIFAR-10 that show that we can estimate the error much better than prior work, which uses the plugin estimator. The key insight is the same as for the $\ell_2$ calibration error—the plugin estimator for the $\ell_1$ calibration error is biased and this bias leads to inaccurate estimates. To estimate the error better we can subtract an approximation of the bias which leads to a better estimate. The main difference is that for the $\ell_1$ calibration error we were not able to approximate the bias with a closed form expression and instead use a Gaussian approximation.

Recall that the $\ell_1$-CE is the $\ell_p$ calibration error with $p = 1$, redefined below:

**Definition G.1.** *The $\ell_1$ calibration error of $f : \mathcal{X} \to [0, 1]$ is given by:*

$$\ell_1\text{-CE}(f) = \mathbb{E}\left[\,|f(X) - \mathbb{E}[Y \mid f(X)]|\,\right] \tag{7}$$

Estimating the $\ell_1$-CE for many models is challenging (see Section 3) so prior work instead selects a binning scheme $\mathcal{B}$ and estimates $\ell_1$-CE$(f_\mathcal{B})$ of a model $f$. Suppose we wish to measure the binned calibration error $\mathcal{E} = \ell_1$-CE$(f_\mathcal{B})$ of a model $f : \mathcal{X} \to [0, 1]$ where $|\mathcal{B}| = B$. Suppose we get an evaluation set $T_n = \{(x_1, y_1), \ldots, (x_n, y_n)\}$. Past work typically estimates the $\ell_1$ calibration error using a plugin estimate for each term:

**Definition G.2** (Plugin estimator for $\ell_1$-CE). *Let $L_k$ denote the data points where the model outputs a prediction in the $k$-th bin of $\mathcal{B}$: $L_k = \{(x_j, y_j) \in T_n \mid f(x_j) \in I_k\}$.*

*Let $\hat{p}_k$ be the estimated probability of $f$ outputting a prediction in the $k$-th bin: $\hat{p}_k = \frac{|L_k|}{n}$.*

*Let $\hat{y}_k$ be the empirical average of $Y$ in the $k$-th bin: $\hat{y}_k = \sum_{x,y \in L_k} \frac{y}{|L_k|}$.*

*Let $\hat{s}_k$ be the empirical average of the model outputs in the $k$-th bin: $\hat{s}_k = \sum_{x,y \in L_k} \frac{f(x)}{|L_k|}$.*

*The plugin estimate for the binned $\ell_1$ calibration error is the weighted squared difference between $\hat{y}_k$ and $\hat{s}_k$:*

$$\hat{\mathcal{E}}_{\text{pl}} = \sum_{k=1}^{B} \hat{p}_k |\hat{s}_k - \hat{y}_k|$$

The plugin estimate is a biased estimate of the binned calibration error. Intuitively, this is because of the absolute value: on any finite samples $s_k$ and $y_k$ will differ and the absolute value of the difference will be positive, even if the population values are the same. More concretely consider a model $f$ where $\mathbb{E}[f(X) \mid f(X) \in I_k] = \mathbb{E}[Y \mid f(X) \in I_k]$ in every bin $k$. In that case the binned calibration error $\mathcal{E}$ is 0. But on any finite samples the plugin estimate $\hat{\mathcal{E}}_{\text{pl}}$ will be larger than 0. In particular, the plugin estimator overestimates the binned calibration error, and the extent of overestimation may be different for different models.

To improve the estimate, we can subtract an approximation of the bias. That is, we would like to output $\hat{\mathcal{E}}_{\text{pl}} - (\mathbb{E}[\hat{\mathcal{E}}_{\text{pl}}] - \mathcal{E})$ as our estimate of the calibration error, where $\mathbb{E}[\hat{\mathcal{E}}_{\text{pl}}] - \mathcal{E}$ is the bias. However, $\mathbb{E}[\hat{\mathcal{E}}_{\text{pl}}] - \mathcal{E}$ is difficult to approximate in closed form. Instead, we propose approximating it by simulating draws from a normal approximation. More precisely, let $y_k = \mathbb{E}[Y \mid f(X) \in I_k]$. Then, each label in the $k$-th bin is a draw from a Bernoulli distribution with parameter $y_k$. So $\hat{y}_k$ is the mean of $n\hat{p}_k$ Bernoulli draws. Assuming that the number of points in each bin is not too small, we can approximate $\hat{y}_k$ using a Gaussian approximation, and use that to approximate the bias $\mathbb{E}[\hat{\mathcal{E}}_{\text{pl}}] - \mathcal{E}$.

**Definition G.3** (Debiased estimator for $\ell_1$-CE). *For each $k$, let $R_k$ be a random variable sampled from a normal approximation of the label distribution in the $k$-th bin: $R_k \sim N(\hat{y}_k, \frac{\hat{y}_k(1-\hat{y}_k)}{n\hat{p}_k})$. The debiased estimate for the binned $\ell_1$ calibration error subtracts an approximation of the bias from the plugin estimate:*

$$\hat{\mathcal{E}}_{\text{db}} = \hat{\mathcal{E}}_{\text{pl}} - (\mathbb{E}\left[\sum_{k=1}^{B} \hat{p}_k |\hat{s}_k - R_k|\right] - \hat{\mathcal{E}}_{\text{pl}})$$

(a) $B = 15$                        (b) $B = 100$

Figure 12: Mean-squared errors of plugin and debiased estimators on a recalibrated VGG16 model on ImageNet with 90% confidence intervals (lower values better). The debiased estimator is closer to the ground truth, which corresponds to 0 on the vertical axis, and much more so when $B$ is large or $n$ is small. Note that this is the MSE of the ECE estimates, not the MSE of the model in Figure 3.

We can approximate the expectation in the debiased estimator by simulating many draws from a normal distribution, which is computationally fairly inexpensive. Note that our proposed estimator is a heuristic approach, and future work should examine whether we can get provably better estimation rates for estimating the $\ell_1$ calibration error, as we did for the $\ell_2$ calibration error. That might involve analyzing our proposed estimator, or may involve coming up with a completely different estimator.

We run multiclass top-label calibration experiment on CIFAR-10 and ImageNet which suggests that the debiased estimator produces better estimates of the calibration error than the plugin estimator. We describe the protocol for ImageNet first, which is similar to the experimental protocol in Section 5.1. We split the validation set of size 50,000 into two sets $S_C$ and $S_E$ of sizes 3,000 and 47,000 respectively. Note that a practitioner would not need so many data points when estimating their model's calibration, we use 47,000 points only so that we can reliably compare the estimators. We use $S_C$ to re-calibrate a trained VGG-16 model and select a binning scheme $\mathcal{B}$ so that each bin contains an equal number of points (uniform-mass binning). We calibrate the top probability prediction as described in Section 2 using Platt scaling. For varying values of $n$, we sample $n$ points with replacement from $S_E$, and estimate the binned $\ell_1$ calibration error (ECE) using the plugin estimator and our proposed debiased estimator. We used $B = 100$ or $B = 15$ bins in our experiments. We then compute the squared deviation of these estimates from the binned $\ell_1$ calibration error measured on the entire set $S_E$. We repeat this resampling 1,000 times to get the mean squared deviation of the estimates from the ground truth and 90% confidence intervals. Figure 12 shows that the debiased estimates are much closer to the ground truth than the plugin estimates—the difference is especially significant when the number of samples $n$ is small or the number of bins $B$ is large. Note that having a perfect estimate corresponds to 0 on the vertical axis.

For CIFAR-10 we use the same protocol except we split the validation set of size 10,000 into two sets $S_C$ and $S_E$ of sizes 3,000 and 7,000 respectively. Figure 4a shows that the debiased estimates are much closer to the ground truth than the plugin estimates in this case as well.

## G.2    Additional experiments for estimating calibration error

In Section 5 we ran an experiment on CIFAR-10 to show that the debiased estimator gives estimates closer to the true calibration error than the plugin estimator. To give more insight into this, Figure 14 shows a histogram of the absolute difference between the estimates and ground truth for the plugin and debiased estimator, over the 1,000 resamples, when we use $B = 10$ or $B = 100$ bins. For $B = 10$ bins it is not completely clear which estimator is doing better but the debiased estimator avoids very bad estimates. However, when $B = 100$, the debiased estimator produces estimates much closer to the ground truth (0 on the x-axis).

(a) $B = 15$             (b) $B = 100$

Figure 13: Mean-squared errors of plugin and debiased estimators on a recalibrated VGG16 model on CIFAR-10 with $90\%$ confidence intervals (lower values better). The debiased estimator is closer to the ground truth, which corresponds to 0 on the vertical axis, especially when $B$ is large or $n$ is small. Note that this is the MSE of the ECE estimates, not the MSE of the model in Figure 3.

(a) $B = 10$ bins             (b) $B = 100$ bins

Figure 14: Histograms of the absolute value of the difference between estimated and ground truth squared calibration errors (0 on the x-axis). For $B = 10$ bins, the results are mixed but we avoid very bad estimates. For $B = 100$ our estimates are much closer to ground truth.

We also show histograms for the ECE experiments in Appendix G.1, in Figure 15 for ImageNet and Figure 16 for CIFAR-10. These histograms show that the proposed debiased estimator produces estimates much closer to the ground truth than the plugin estimator.

We also ran a marginal multiclass calibration experiment on CIFAR-10 to show that our estimator allows us to select models with a lower mean-squared error subject to a given calibration constraint. In this case we split the validation set into $S_C$ and $S_E$ of size 6000 and 4000 respectively, and recalibrated a trained model on $S_C$. On $S_E$, we estimate the calibration error using the plugin and debiased estimators and use 100 Bootstrap resamples to compute a 90% upper confidence bound on the estimate (from the variance of the Bootstrap samples). We compute the mean-squared error and the upper bounds on the calibration error for $B = 10, 15, \ldots, 100$ and show the Pareto curve in Figure 17. Figure 17 shows that for any desired calibration error, the debiased estimator enables us to pick out models with a better mean-squared error. For example, if we want a model with calibration error less than $1.5\%$, the debiased estimator tells us we can confidently use 100 bins, while relying on the plugin estimator only lets us use 15 bins and incurs a 13% higher mean-squared error.

(a) $B = 15$ bins

(b) $B = 100$ bins

Figure 15: Histograms of the absolute value of the difference between estimated and ground truth ECE (0 on the x-axis) on ImageNet.

(a) $B = 15$ bins

(b) $B = 100$ bins

Figure 16: Histograms of the absolute value of the difference between estimated and ground truth ECE (0 on the x-axis) on CIFAR-10.

Figure 17: Plot of mean-squared error against 90% upper bounds on the calibration error computed by the debiased estimator and the plugin estimator, when we vary the number of bins $B$. For a given calibration error, our estimator enables us to choose models with a better mean-squared error. If we want a model with calibration error less than 0.015, the debiased estimator tells us we can confidently use 100 bins, while relying on the plugin estimator only lets us use 15 bins and incurs a 13% higher mean-squared error.