[Reviews · NeurIPS 2019]

Reviewer 1



This paper adresses the problem of calibration in predictive models in two ways: providing a method for obtaining sample efficient calibration models and showing some properties of estimates for calibration errors known in the literature. Major issues: - (Meaning of variance-reduced) The denotation of your method as variance-reduced does not convince me, as the reduction of the variance refers only to the histogramm binning approach, if I understand it correctly. Which, on the other hand, is due to the fitting of the scaling function in the first step of your method. Thus, scaling methods could also be referred to as variance-reduced methods, couldn't they? Please correct me, if I am wrong. - (Theorem 4.1) Here, the statement is more or less: your method is almost as well-callibrated as the best possible recalibrator among the recalibrator set G after a certain number of samples. I understand your message that the calibration error of your technique can be estimated, while this is usually not the case for scaling methods. But wouldn't it nevertheless be better to use the scaling method for the calibration, while your variance-reduced method serves as a surrogate in order to check if the error is smaller than some threshold? Since if the estimated calibration error of the variance-reduced calibrator is with high probability smaller than some threshold, the calibration error of the underlying scaling method is smaller than this threshold as well. This would address the issue of scaling methods you mention in line 126. - (Proof of Theorem 4.1) Fist, your n in line 575 must be of order 1/epsilon^4 in order to derive the statement in line 576 by means of Lemma D.1. But this would imply a sample size of a higher order than you state in the assertion of the theorem. Did you initially want to prove an epsilon bound instead of an epsilon^2 bound? This would be in accordance with your line 287. Second, you have in Lemma D.3 a high probability statement (also in Lemma D.1), but in your application of Lemma D.3 in lines 578-579 you do not mention this high probability statement anymore. Therefore, your statement of the theorem is a high probability statement as well, which, on the other hand, is missing in the formulation of the theorem. In summary, the paper suggests an intuitive approach to the problem of calibrating prediction models by binning the output of a fitted scaling function. This approach seems reasonable, and in addition, the authors show that it comes with many benefits such as sample efficiency and the possibility of estimating the calibration error in contrast to other scaling methods. Furthermore, in the experimental studies the superiority of their suggested calibration technique is shown over the popular histogramm binning. However, as mentioned above, I believe that the method is better suited to access the calibration error of the underlying scaling method due to Theorem 4.1. Maybe it is possible to compare the method with a scaling method on some synthetic data scenario, where one has control over the distributions of X (resp. Z) and Y, and where one is able to derive the calibration error of the scaling method in closed form or approximate it appropriately. This could give some insights on the deviation of the method's calibration error to the calibration error of the scaling method. Nevertheless, I think the paper provides some valuable contributions to the considered field of research: First, the sample complexity for estimates of the calibration error and, second, revealing the issue of underestimation of the calibration error for scaling methods. Minor issues: - In each mathdisplay the period at the end is missing. - line 74: You have to use the p-th power of the absolute value of the difference. - lines 90 and line 98: It should be f instead of M respectively. - lines 115 and 183: the class G should be specified. - line 118,119: The transition is somehow abrupt. Maybe write this passage more coherent. - line 119: It should be I_1,..,I_B instead of I_1,..,I_j. - line 122: Here, g denotes the histogramm binning itself, doesn't it? Maybe be more specific here on g. - Histogramm binning: Considering also the last point, I think it would be helpful to give a similar definition of the histogramm binning as in line 190 for the variance-reduced calibrator. - line 189: "... mean of a set of values S." - line 195: Here, you switch to a caligraphical G for the class of recalibrators. - line 214: "In Section 3 we showed ..." is somehow exaggerated: you rather have argued that this is the case in lines 130 - 133. - Definition 4.2: It should be \alpha >=1. Also you have not defined Z. I guess this is just Z=f(X). - line 224: It should be Lemma 4.3 instead of Lemma D.4. And in the supplement it should be consequently Lemma 4.3 instead of Lemma D.4. - line 260: "... measure the calibration error of binned model ...". - line 265: \hat y_S instead of \hat y_i. - line 270: you have not defined E^* before. - Theorem 5.3 and 5.4: Is the upper bound of the 2-well-balanced property not needed? - line 450: has 'a' density. - line 479: a period is missing at the end. - Proof of Lemma D.1: again you have not defined Z (only in the proof of Prop.3.3 in line 466). - lines 509-510: it should be the difference in the calibration terms. - Definition D.2: Why do you state the L_2 norm after the squared L_2 norm? This is clear that it is the square root of the squared L_2 norm. After rebuttal: The authors' response clarified the most important issues, in particular the way in which the theorem should be understood. Also, the adaptation of the theorem appears to be appropriate. Overall, I would now also opt for accepting the paper. Minor: The effect of delta on the sample complexity is still a bit unclear. Moreover, I'm still not fully convinced by the name of the method.

Reviewer 2



1. Originality. As some of the definitions of calibration error, plugin estimator and debiased estimator, etc. were already proposed before, the authors' main contribution lies in their putting the existing works and definitions together to form a more complete statistical framework for estimating and evaluating the calibration error. The other contribution is that the authors propose a new calibration method by first fitting a scaling function and then binning the scaling function outputs and prove that it has better properties in terms of sample complexity and measurability. 2. Quality. The claims in the paper are validated by experiments on 2 image classification datasets, CIFAR-10 and ImageNet. The authors provide code for reproducibility. One of my concerns is that the authors claim that the theory is presented for binary classification setting, while the experiments are performed on multi-class classification. It is better to conduct experiments on binary classification as well to validate the error bounds given by the theorems and extend the theorems to multi-class settings. 3. Clarity. 3.1 The notations are confusing and contain some typos. The authors should give a clear definition first for the notations used in the theorem/definition. Definition 2.2-2.3 should state clearly the expectation is taken over which distribution. M(X) used in Definition 2.2 and 2.3 is not defined. Definition 2.3, in the bracket should be top-label calibration error Definition 2.4 P(j) should be P(k) in my understanding Definition 3.1 misses a term that the authors want to define. It is better to define I_j again. Definition 4.2, it should be alpha >= 1. 4. Significance. The authors provide a more complete statistical framework for estimating and evaluating the recalibration error. The theorems and the methods are likely to be used and compared by future researchers working on uncertainty calibration. However, one of the limitations is that this framework requires the well-balanced binning property holds, but this may not always be satisfied on real data.

Reviewer 3



# update after author feedback I have read all other reviews and author feedback. The feedback partly satisfies my request for greater clarity regarding asymptotics notation; on line 178 I now understand the subtlety and would recommend either clarifying or not attempting to push this as an important claim. The new formulation of thm 4.1 makes more sense now. It's good to have the experimental results requested by R1 and R2 as they support the conclusions. I second R1 in that the name you give to your method sounds like a misnomer to me. It might be worth finding a better name if this is to be published. I don't think I can increase my score still as the next notch is quite demanding (top 15% of accepted contributions), instead I stay firm on my current score. # original review ## originality The main algorithmic contribution (a new recalibration procedure) is a composition of two existing methods (Platt scaling, histogram binning). The combination is not necessarily original, but happens to bring substantial sample efficiency advantages which justify its importance. ## quality The analysis is exceptionally thorough. This paper summarises a large body of work. ## clarity The paper is exceptionally clear. The setup sec2 is careful and clear, also covers the multiclass case accurately, and well illustrated by the simple example in fig1 which is used throughout. The use of a "mathematical presention" with theorems, lemmas, definitions etc is not merely an artifice, but serves to clarify. ## significance As argued in the analysis of contributions, I believe this paper is of high practical significance.

Reviewer 4



1. The contributions of this paper (summarized above) are original, useful and clearly presented. 2. The new 'variance-reduced' calibration approach is an interesting proposal. It uses a traditional 'scaling' approach to obtain recalibrated forecasts, groups these forecasts in to bins, and then replaces each forecast with the average of the forecasts in the same bin. This new approach is worse than scaling alone in two important ways: the overall performance (as measured by mean squared error) is slightly degraded (appendix D.2), and the new approach is less efficient (as measured by the number of samples needed to achieve a given level of calibration error: lines 174 and 193). The advantage of the new approach is that its calibration error can be estimated more accurately. This advantage is useful if forecasts are chosen to optimize overall performance subject to a limit on the calibration error (appendix G). This criterion reflects the commonly stated goal of probability forecasting to optimize performance subject to calibration but I've not seen the criterion used in practice, except in this paper. Much more common, in my experience, is for forecasts to be chosen simply to optimize overall performance, and this might favour scaling alone over the new approach. The paper would benefit from adding a stronger argument for optimizing performance subject to a calibration budget. For example, it would be useful to discuss at greater length why a calibration budget might be important and how it might be set in practice. 3. The paper criticizes scaling methods because their calibration error is typically under-estimated. The method used to estimate the calibration error in Figure 2 uses a simple 'plug-in' estimator. Another 'debiased' estimator is discussed in section 5. This latter estimator is designed for discrete probability forecasts, but I wonder what results it would give if it were applied to the binned forecasts used to produce Figure 2. If the resulting estimates of calibration error were less sensitive to the number of bins then this might encourage the use of scaling approaches even when a calibration budget is imposed. The paper would be stronger if it investigated this point. Minor comments Equation 2: Define M(X)_j. Equation 3: Replace P(j) with P(k) and replace the second expectation with probability. Definition 2.4: Define P(j) more precisely. Is it the probability of class j occurring among the labels? Definition 2.4: The decision to weight the calibration error for each class by P(k) seems like a reasonable choice to me, but I think it is still a choice that deserves a (brief) justification. Line 122: Replace g(z_i) with z_i? Line 244: As the mean squared error is better for the same calibration error, this must mean that the sharpness (line 83) is better. Consider adding a (brief) note explaining how the variance-reduced approach achieves that.

[Author Response · NeurIPS 2019]

We thank all the reviewers for their thorough reviews. All reviews expressed that there were "valuable contributions" in
the paper, and R3 and R4 said the work was of "high practical significance" and "original, useful and clearly presented".
The reviewers also had many constructive suggestions and questions that we will address below.

**R1, scaling vs ours**: R1's first major question was if, due to Theorem 4.1, one should "use the scaling method for
calibration" and use our method "as a surrogate in order to check if the (scaling method's) error is smaller than some
threshold". We cannot do this because the calibration error of the scaling method can be much higher than for our
method. Theorem 4.1 says that our method is *at least* "almost as well-calibrated as the best possible recalibrator" in "G
after a certain number of samples", but it could be much better calibrated, as in Example 3.2. This is because the binned
version of a function has lower calibration error than the original function (Proposition 3.3, used in line 206 of Theorem
4.1 proof sketch). This is a fundamental issue with scaling methods—binning only lower bounds their calibration error.

**R1, additional experiment**: As suggested, we ran synthetic experiments to compare our calibrator with the underlying
scaling method. The ground truth $P(Y = 1|z)$ is from the Platt scaling family $G$ but with noise. Varying $n$, we compute
90% confidence intervals from 1000 trials. With 10 bins, $n = 3000$ the $\ell_2^2$ calibration error is $5.2 \pm 1.1$ times lower
for our method than the scaling method—our method does even better for larger $n$. And unlike scaling methods, our
method has measurable calibration error—if we are not calibrated we can get more data or use a different scaling family.

**R1, proof of Theorem 4.1**: $R1$ mentioned that to use Lemma D.1, we would in fact need $1/\epsilon^4$ points to achieve
$\ell_2^2$-CE $\le 2\epsilon^2$. We made a mistake (thanks for catching it), but it can be easily repaired as follows. In lemma D.1,
we can actually get a convergence rate of $1/n$ instead of $1/\sqrt{n}$ for the MSE, using standard asymptotic results of
M-estimators (under regularity conditions). Technical details: The asymptotic result gives a *parameter* convergence
rate of $1/\sqrt{n}$ which leads to a $1/n$ convergence rate in the MSE *loss*. We have updated Lemma D.1, which fixes line
575. We have fixed the theorem statement (see below) and applications of the lemmas to clarify they are probabilistic.
We also implemented synthetic experiments to sanity check these bounds (see 'R2, validating bounds').

**R1, other concerns**: We agree with all of R1's detailed comments and will fix them (for example we have toned down
line 214 to say "we showed that *current techniques* cannot accurately measure the calibration error of scaling methods").

**R2** had 2 main concerns. 1. **Well-balanced binning**: R2 was concerned that our framework requires well-balanced
binning to hold, which may not hold on real data. We believe this is a misunderstanding—in step 2 of our algorithm we
*choose* bins so that an equal number of calibration points land in each bin. We then *prove* (instead of require) that the
well-balanced property holds in the population (Lemma 4.3). 2. **Binary vs Multi-class**: Our theory generalizes to the
multiclass setting. Top-label calibration is a binary calibration problem (lines 103 - 104) where $Z \in [0, 1]$ is the model's
confidence for the top class, and $Y$ is 1 if the model's prediction was correct, and 0 otherwise. Marginal calibration
requires each class to be independently calibrated, which transforms into $K$ binary calibration problems where $K$ is the
number of classes. **Notational issues and typos**: We thank the reviewer for identifying them, and will fix these.

**R2, validating bounds**: As R2 suggested, we added synthetic experiments to validate the bound in Theorem 4.1. Our
theory predicts that $n \lesssim 1/\epsilon^2 + B$ for our method but for histogram binning $n \lesssim B/\epsilon^2$. In the first experiment, we fix
$B$ and vary $n$—we see that $1/\epsilon^2$ is approximately linear in $n$ for both calibrators. In the second experiment, we fix $n$
and vary $B$—as predicted by the theory, for our variance-reduced calibrator $1/\epsilon^2$ is nearly constant, but for histogram
binning $1/\epsilon^2$ scales close to $1/B$. When we increase from 5 to 20 bins, our method's $\ell_2^2$-CE decreased by $2\% \pm 7\%$ but
for histogram binning it increased by $3.71 \pm 0.15$ *times*—we will include details and plots in the paper.

**R3** had a number of useful suggestions and questions. They mentioned that the use of big-O in Theorem 4.1 was
confusing—we have rephrased the theorem as shown below. We agree that the DEMOGEN dataset is a good resource
to tap into for a more extensive analysis of calibration—we will mention this as potential future work and cite the
dataset/paper. Regarding line 178: histogram binning bins the $Y$ values, but not the outputs of a *recalibrator function*.
We will address R3's other suggestions (e.g. connection with scoring rules) in the next revision.

**Theorem 4.1**: Assume regularity conditions on $\mathcal{G}$ (Lipschitz, injective, and all conditions in Theorem 5.23 in Asymptotic
Statistics, Vaart, A.) Given $\delta \in (0, 1)$, there is a constant $c$ such that *for all $B, \epsilon$, with $n \ge c\left(B \log B + \frac{\log B}{\epsilon^2}\right)$ samples,*
the variance-reduced algorithm finds $\hat{g}_{\mathcal{B}}$ with $\ell_2^2$-CE$(\hat{g}_{\mathcal{B}}) \le \min_{g \in \mathcal{G}} \ell_2^2$-CE$(g) + \epsilon^2$, with probability $\ge 1 - \delta$.

**R4** had a good suggestion—checking if the debiased estimator was less sensitive to the number of bins when used
for scaling methods. We repeated the experiment in Section 3.1, but observed similar results, which we will add to
the paper. Regarding motivation, we believe practitioners use calibration in many ways although [5] (Gneiting and
Raftery, Science 2005) propose maximizing sharpness subject to a calibration error budget. We believe practitioners
implicitly do this—calibration/reliability is an important diagnostic metric and when it is unsatisfactory the forecast and
its granularity are changed. R4 mentioned that our method is less efficient than scaling methods—note that our method
typically has lower (better) calibration error than the scaling method, as binning decreases the calibration error, e.g. see
"R1, additional experiment". We thank R4 for the detailed comments and will fix the issues identified.

[Meta-Review · NeurIPS 2019]

Congratulations, your paper has been accepted to NeurIPS2019. The reviewers found it to be original, important, and well written. When preparing the camera ready version, please bear in mind the reviewers comments. In particular, the following were raised during the discussion as outstanding points that should be considered when revising the paper: - discussion of the idea of a calibration budget - this is important because, whereas the new approach might improve calibration relative to traditional scaling, it tends to worsen sharpness (because overall performance is slightly degraded - appendix D.2). Without a discussion of the relative importance of overall performance, calibration and sharpness, and without highlighting the impact of the new approach on *all* of these attributes (compared to traditional scaling), readers might get the impression that there is little downside to the new approach. - Include the clarification from your response of the way in which theorem 4.1 should be understood. - The effect of delta on the sample complexity is still a bit unclear - can this be made clearer? - Include the information clarifying the asymptotics notation from your response. The follow on line 178 I now understand the subtlety and would recommend either clarifying or not attempting to push this as an important claim. - Include the new experimental results requested by R1 and R2 (that were in your response), as they support the conclusions. - There was a feeling that the name given to your method is misleading - is it worth finding a more appropriate name for your approach?